# Sensitivity of air quality model responses to emission changes: comparison of results based on four EU inventories through FAIRMODE benchmarking methodology.

5    Alexander de Meij[1], Cornelis Cuvelier[2♦], Philippe Thunis[2], Enrico Pisoni[2], Bertrand Bessagnet[2]

[1]MetClim, Varese, 21025, Italy

[2] European Commission, Joint Research Centre (JRC), 21027, Ispra, Italy

♦retired with Active Senior Agreement

*Correspondance to* : Philippe Thunis (philippe.thunis@ec.europa.eu)

**Abstract**

Despite the application of an increasingly strict EU air quality legislation, air quality remains problematic in large parts of Europe. To support the abatement of these remaining problems, a better understanding of the potential impacts of emission abatement measures on air quality is required and air chemistry transport models (CTMs) are the main instrument to perform emission reduction scenarios. In this study, we study the robustness of the model responses to emission reductions when emission input is changed. We investigate how inconsistencies in emissions impact the modelling responses in the case of emission reduction scenarios. Based on EMEP simulations over Europe fed by four emission inventories: EDGAR 5.0, EMEP-GNFR, CAMS 2.2.1 and CAMS version 4.2 (incl. condensables), we reduce anthropogenic emissions in six cities (Brussels, Madrid, Rome, Bucharest, Berlin and Stockholm) and 2 regions (Po Valley Italy and Malopolska Poland) and study the variability of the concentration reductions obtained with these four emission inventories.

Our study reveals that the impact of reducing aerosol precursors on PM10 concentrations result in different potentials and potencies, differences that are mainly explained by differences in emission quantities, differences in their spatial distributions as well as in their sector allocation. In general, the variability among models is larger for concentration changes (potentials) than for absolute concentrations. Similar total precursor emissions can however hide large variations in sectorial allocation that can lead to large impacts on potency given their different vertical distribution. PPM appears to be the precursor leading to the major differences in terms of potentials. From an emission inventory viewpoint, this work indicates that the most efficient actions to improve the robustness of the modelling responses to emission changes would be to better assess the sectorial share and total quantities of PPM emissions. From a modelling point of view, NOx responses are the more challenging and require caution because of their non-linearity. For O3, we find the relationship between emission reduction and O3 concentration change shows the largest non-linearity for NOx (concentration increase) and a quasi-linear behaviour for VOC (concentration decrease).

We also emphasize the importance of accurate ratios of emitted precursors since these lead to changes of chemical regimes, directly affecting the responses of O3 or PM10 concentrations to emission reductions.

## 1. Introduction

Despite the application of an increasingly strict EU air quality legislation, air quality remains problematic in large parts of Europe (EEA, 2020). This becomes even more crucial now that more stringent recent WHO guideline values (WHO, 2021) as well as the recently proposed EU limit values (EC, 2022) have acknowledged that air pollution can have negative impacts on health at much lower concentration levels for air pollutants such a PM10, PM2.5 and NOx. To comply with these higher-ambition limit values, a better understanding of the potential impacts of emission abatement measures on air quality is required. Air chemistry transport models (CTMs) are the main instrument to perform emission reduction scenarios, helping scientists and policymakers to understand which and how much of the emissions should be reduced to improve air quality. Over the years, CTMs continuously evolved by implementing more exhaustive and detailed chemical and dynamical atmospheric processes and higher spatial grid resolution to capture fine-scale features driven by land surface characteristics (De Meij et al., 2015, 2018).

Many studies exist that analyse the sensitivity of baseline concentrations to emissions or have compared model responses among themselves (Thunis et al., 2007, 2010, 2013, 2021a Vautard et al., 2007, Mircea et al., 2019). To the knowledge of the authors, very few works assessed the sensitivity of model responses to the emission input, e.g. De Meij et al. (2009), Aman et al., (2011) Miranda et al. (2015 and references therein). Other studies have investigated the uncertainties associated with certain processes when air chemistry models are used to support policy making, such as meteorological input (De Meij et al., 2009a and references therein), aerosol chemistry (Thunis et al., 2021a, Clappier et al., 2021), model resolution (De Meij et al., 2007), or the emissions (Thunis et al., 2021b and references therein). Many of these topics are addressed in the frame of the Forum for Air Quality Modelling (FAIRMODE) (https://fairmode.jrc.ec.europa.eu/home/index) that provides air quality modellers with a permanent forum to address air quality modelling issues. One of FAIRMODE's goals is also to assess the sensitivity of model responses to emission reductions in general. In this study, the robustness of the model responses to emission reductions is assessed when the emission input data are changed. While in Thunis et al. (2022), the authors compared emission inventories among themselves and proposed an approach to identify inconsistencies, we here investigate how these inconsistencies impact the modelling responses in the case of emission reduction scenarios. It is indeed crucial to better assess the share of the uncertainty that is associated to emission inventories in the overall uncertainty of the modelling response (Georgiou et al., 2020) as this is a key model output when designing air quality plans.

In the light of the above, we investigate in this work the robustness of model responses to emission changes with a CTM based on four emission inventories and use specific indicators for the analysis. To this end, we perform simulations over Europe with the air chemistry transport model EMEP (Simpson et al., 2012), fed by the EDGAR 5.0, EMEP-GNFR, CAMS version 2.2.1 and CAMS version 4.2 + condensables emission inventories. We reduce anthropogenic emissions in six cities (Brussels, Madrid, Rome, Bucharest, Berlin and Stockholm) and 2 regions (Po Valley Italy and Malopolska Poland) and study the variability of the concentration reductions obtained with these four emission inventories feeding the EMEP model, considering a meteorology fixed at 2015. More details

on the model, methodology and emission inventories are given in Chapter 2. We discuss the results in Chapter 3 and we conclude in Chapter 4.

## 2.     Methodology

Four emission inventories are used to feed the EMEP model to understand how this input data influences the calculated model changes in air pollutant concentrations. We performed one BaseCase (no emission reduction) simulation with each emission inventory for the year 2015 over Europe.

For the scenarios, we reduced for each emission inventory, the emissions of NOx, VOCs, NH3, SOx and primary particulate matter (PPM which includes both their fine (size <2.5 μm) and coarse (2.5 μm< size <10 μm) by 25% and 50% for each species separately. This is done for six cities (Brussels, Madrid, Rome, Bucharest, Berlin and Stockholm) and two regions (Malopolska, Poland and Po Valley, Italy) to study the impact on particulate matter (PM) and ozone (O3) formation. More details on the model and the emission inventories are given in the next section. Because emissions are reduced in all cities/regions in a single simulation, these cities/regions must be far away from each other to avoid that emission reductions applied in one location influences background concentration levels in others . This constraint limits the number of cities/regions that we can cover in this work. These emission reductions are theoretical and do not link with  specific measures. For Malopolska and the Po Valley the emissions are reduced over the whole modelling domain, as described in Table S1 of the Electronic Supplement. However, we analyse the impact of the emission reductions only over the city centres of Krakow and Milan, respectively. An overview of the characteristics of each modelling domain and the area over which the emissions are reduced is provided in Table S1 (ES). Below we present the air quality model and the emission inventories used in this study, together with the relevant indicators considered for this study.

### 2.1     The EMEP air quality model

In this study the EMEP model version rv_34 is used, which is an off-line regional transport chemistry model (Simpson et al., 2012; https://github.com/metno/emep-ctm), to study the sensitivity of model responses to emission changes.

The model domain stretches from -15.05° W to 36.95° E longitude and 30.05° N to 71.45° N latitude with a horizontal resolution of 0.1° x 0.1° and 20 vertical levels, with the first level around 45 m. The EMEP model uses meteorological initial conditions and lateral boundary conditions from the European Centre for Medium Range Weather Forecasting (ECMWF-IFS) for the meteorological year 2015. The temporal resolution of the meteorological input data is daily, with 3-hours timesteps. The initial and background concentrations for ozone are based on Logan (1998) climatology, as described in Simpson et al. (2003). For the other species, background/initial conditions are set within the model using functions based on observations (Simpson et al., 2003 and Fagerli et al., 2004). Secondary aerosol formation (Simpson et al., 2012)) accounts for  complex chemical and physical processes , such as sulphate aerosol formation from SO2, nitrate aerosol formation from NOx or organic

aerosol formation from VOCs. More detailed information on the meteorological driver, land cover, model physics and chemistry are provided in De Meij et al., (2022) and references therein.

## 2.2 Emission inventories

In this study we used the anthropogenic emissions of four emission inventories, all for the year 2015. The emission inventories are:

    1. EDGAR v5.0.
    2. EMEP-GNFR
    3. CAMS-REG v2.2.1.
4. CAMS-REG v4.2 with condensables.

Note that, while EDGAR is completely independent from the other emission inventories, there are common features in the other three inventories. For example, emission inventories 2 and 3 share the same country totals but use different proxies to spatialize emissions; while emission inventories 3 and 4 differ in terms of release date and emission updates from 2.2.1 to 4.2 with 4 also containing condensables in addition to 3. "Condensables"

represent the fraction consisting of organic vapour able to react and/or produce condensed species when cooling.

All emissions are detailed in terms of the GNFR classification (Table 2 of the ES, where GNFR stands for Gridded Nomenclature For Report. An overview of the characteristics of the emissions inventories is given in Table 1. The anthropogenic emissions in the four inventories are: CO, NOx, SOx, NH3, VOC, PM25, PM10.

### 2.2.1 EDGAR

The Emissions Database for Global Atmospheric Research (EDGAR) is a global inventory providing greenhouse gas and air pollutant emissions estimates for all countries over the time period 1970 till most recent years, covering

all IPCC reporting categories, with the exception of Land Use, Land Use Change and Forestry (LULUCF). It uses a bottom-up approach, i.e. using activity data and country specific emission factors based on IPCC recommendations to estimate emission quantities (Crippa et al., 2018).

For this work, we use the EDGAR 5.0 inventory (further denoted as EDGAR), that contains anthropogenic emissions for aerosol and aerosol precursor gases at 0.1 x 0.1 horizontal resolution. The inventory is available at

https://EDGAR.jrc.ec.europa.eu/dataset_ap50; Janssens-Maenhout et al., 2019, Crippa et al., 2020). More information about the emission inventory is given in Thunis et al., 2021b and references therein.

### 2.2.2 EMEP-GNFR

The EMEP emissions (Mareckova et al., 2017), further denoted as EMEP-GNFR, are compiled within the

"UNECE co-operative programme for monitoring and evaluation of the long-range transmission of air pollutants in Europe" (unofficially 'European Monitoring and Evaluation Programme', EMEP). EMEP is a scientifically based and policy driven programme under the Convention on Long-range Transboundary Air Pollution

(CLRTAP) for international co-operation, that has the final aim of solving transboundary air pollution problems. More specifically, the EMEP emissions are built from officially reported data provided to CEIP (Centre of Emission Inventory and Projection, a body of EMEP) by the Convention Parties. Emissions are gap-filled with gridded TNO data from Copernicus Atmospheric Monitoring Service (CAMS) and EDGAR. The dataset consists of gridded emissions for SOx, NOx, NMVOC, NH3, CO, PM2.5, PM10 and PMcoarse at 0.1° x 0.1° resolution. More information on the emissions and where to download can be found in the User Guide (https://emep-ctm.readthedocs.io/_/downloads/en/latest/pdf/) and in Mareckova et al., (2017).

### 2.2.3    CAMS-REG v2.2.1

The Copernicus Atmosphere Monitoring Service Regional Anthropogenic Air Pollutants (CAMS-REG-AP) emission inventory (Granier et al., 2019) covers emissions for the UNECE-Europe for CH4, NMVOC, CO, SO2, NOx, NH3, PM10, PM2.5 and CO2 and CH4. Version 2.2 (further denoted as CAMS221) and newer is an update of the TNO_MACC, TNO_MACC-II and TNO_MACC-III inventories (Kuenen et al., 2014, 2021).

The CAMS-REG-AP methodology starts from the emissions reported by European countries to UNFCCC (for greenhouse gases) and to EMEP/CEIP (for air pollutants), aggregated into different combinations of sectors and fuels. Then, these emissions are gridded using ad-hoc proxies, that differ from the ones used in EMEP-GNFR . The spatial resolution of the emissions is 0.1° x 0.05°. More information can be found in Granier et al. (2019) and Thunis et al., (2021b).

### 2.2.4    CAMS-REG v4.2 + condensables

This inventory (Kuenen et al., 2021, 2022) is an update of the previous CAMS versions for PM emissions for the residential sector, also known as REF1, in which PM2.5 and PM10 emissions have been updated with information on the condensable part (personal communication J. Kuenen, TNO, 2021). This inventory, also known as REF2, is hereafter denoted as CAMS42C. Condensables replace country reported PM2.5 and PM10, with a bottom-up estimate for small combustion for all fuels (not only wood but also for fossil fuels). Since 2016, more and more countries gradually included condensable emissions of small combustion devices, leading to significant differences as shown by Kuenen et al. (2022). For example, in countries such as Poland and Turkey where coal combustion in households is still an important contributor to PM, large emissions of fine and coarse condensables (118kTons/year for PM25) still take place. For Turkey the difference in PM2.5 emissions for GNFR Sector 03 is around 20% (higher in CAMS42C). For Hungary, Slovakia, Ireland, UK, Belgium and Norway the PM2.5 emissions for GNFR Sector 03 are in general lower than in CAMS42C.

Edgar uses a bottom-up approach for all emission source sectors, based on estimates of activity data and emission factors whereas CAMS is mainly based on countries reported emissions. The differences between the same years between the CAMS inventories stems from the recalculations of the pollutants for each country.

More in-depth analysis and explanations on the underlying differences between the emission inventories – as used in this study - is given in Thunis et al. (2022). They identify the largest inconsistencies between the emission inventories in terms of pollutant and sector for 150 cities in Europe and show that the difference for some air

pollutants between emission inventories can be as large as a factor of 100 or more. They explain that the underlying reason for these discrepancies is related to the differences in spatial proxies, country totals (i.e. differences in urban area share) and country sectoral share (e.g. industry, residential, power plants).

**2.3     Indicators for the comparison**

In this section we analyse the impact of the emission reduction on simulated yearly change of concentrations for the six cities and two regions. To perform this analysis, we use the potency and potential indicators as defined in Thunis et al. (2015a, b) based on 50% emission reduction strengths. These indicators are specifically designed to analyse the impact of emission reductions on concentration changes. We only recall their basic definitions here:


The Absolute Potential (AP) is defined as the concentration change (between the BaseCase and the scenario) divided by the reduction strength. It is expressed in µg/m3.

$$AP = \frac{C_{Scenario} - C_{BaseCase}}{\alpha}$$


$C_{BaseCase}$ represents the BaseCase yearly concentrations, obtained with one of the four emission inventories (no emission reduction). $C_{scenario}$ the 'scenario' yearly concentrations and alpha the emission reduction strength, i.e. alpha =0.25 (25% reduction), alpha = 0.50 (50% reduction). All indicators are calculated as 95th percentiles, i.e.

based on the average of all BaseCase concentration values modelled in a given area that exceed the 95th percentile concentration threshold. Note that the grid cells for these concentration values are selected from the BaseCase obtained with a given inventory. They are kept unchanged for the scenario but can differ from one emission inventory to the other. The absolute potential informs on the concentration change projected linearly to 100% from a given scenario.

The relative potential (RP) is obtained by dividing the absolute potential by the BaseCase concentration.

$$RP = \frac{C_{Scenario} - C_{BaseCase}}{\alpha \, C_{BaseCase}}$$

The RP provides similar information as the AP, but because it normalises the concentration change by the BaseCase concentration, it removes the impact of potential biases among BaseCases when different models (here

intended as single model fed by different emissions) are compared to each other.

The Potency (P) in µg/m3/(ton/day) is defined as the ratio of the concentration change by the emission change E.

$$P = \frac{C_{Scenario} - C_{BaseCase}}{E_{Scenario} - E_{BaseCase}}$$


The Potency informs on the potential concentration change per unit emission change. The normalisation by the emission change allows (at least partly) to exclude the impact of differences in the absolute levels of emissions in models when performing the comparison.


### 2.4    Screening method statistical analysis

In this section, we provide a summary of the screening method which is adapted from Thunis et al. (2022). The approach aims at comparing the modelling responses from different models over a series of geographical areas. Based on emissions detailed in terms of precursors (denoted as "p") and city areas (denoted as "c"), the consistency

between two modelled responses (or absolute potential - AP) is decomposed into two aspects: the potency (P) and the underlying emissions (E). To do this, we decompose the ratio of the known absolute potentials of two models for each city as follows:

$$\frac{AP_{p,c}^1}{AP_{p,c}^2} = \frac{\frac{AP_{p,c}^1}{\alpha E_{p,c}^1}}{\frac{AP_{p,c}^2}{\alpha E_{p,c}^2}} * \frac{\alpha E_{p,c}^1}{\alpha E_{p,c}^2} = \frac{P_{p,c}^1}{P_{p,c}^2} * \frac{E_{p,c}^1}{E_{p,c}^2} \tag{1}$$

Superscripts refer to the two models. Equation (1) is an identity where all terms are known from input quantities, i.e. the two modelled absolute potentials detailed in terms of precursors and cities on the left-hand side and the ratios of the potencies and emissions on the right-hand side. E is here intended as the absolute emission values. Multiplied by alpha, we then obtain the emission reduction change, i.e. deltaE=alpha*E.

For convenience, we rewrite equation (1) in logarithm form (2) considering the absolute values of the potencies only, as:

$$log\left(\frac{AP_{p,c}^1}{AP_{p,c}^2}\right) = log\left(\mid\frac{P_{p,c}^1}{P_{p,c}^2}\mid\right) + log\left(\frac{E_{p,c}^1}{E_{p,c}^2}\right) \tag{2}$$

Which can be rewritten as equation (3) with simplified notations:

$$\widehat{AP}_{p,c} = \hat{P}_{p,c} + \hat{E}_{p,c} \tag{3}$$


where the hat symbol indicates that quantities are expressed as logarithmic ratios. These quantities are at the basis of the screening methodology and serve as input for the graphical representation as well. The implicit assumption is that AP1 and AP2 or P1 and P2 have the same sign. This is the case in most cases except in a case of strong non linearities as for Ozone.


We proceed with a number of steps that help focusing on priority aspects. First, we restrict the screening only to absolute potentials that are relevant, i.e. large enough. This is achieved by imposing that any given potential fulfils the condition: $AP_{p,c} > \gamma \times \max\{AP_{p,c}\}$ to be further considered in the screening, where $\gamma$ is a user defined threshold parameter, set to 20% in this work. Second, we flag, among the remaining potentials, only those for

which differences between models are larger than a threshold, β, also set to 20% in this analysis. Beyond this threshold, differences are thought to be large enough to justify further checking. These thresholds are arbitrary

but they should be set in such a way that significant model differences only are spotted while keeping the analysis reasonable (numbers of identified inconsistencies). These thresholds can also be lowered with time if inconsistencies are progressively resolved or explained.


Relation (3) is the basis of the "diamond" diagram (see Fig. 3 as an example) that provides an overview of all inconsistencies detected during the screening process. In this diagram, each inconsistent potential is represented by a point that has absolute total emissions ($\hat{E}$) as abscissa and potency as ordinate ($\hat{P}$). The sum of these two terms ($\widehat{AP}$) is equal for points that lie on "−1" slope diagonals. At this stage it is important to note that positive

differences in terms of emissions and potencies will characterize points lying on the right and top parts of the diagram, respectively. In addition, the upper right and lower left diagram areas indicate summing-up effects whereas the lower right and top left areas highlight compensating effects.

The diamond shape (in the middle of the diagram) derives from equation (3) where the $\beta$ threshold is used to draw

the inconsistency limit for each of its two terms, as well as their sum. Each [p,c] point lying outside this shape is therefore characterized by an inconsistency in terms of either E or P or/and AP, small or large according to its distance from the diamond.

In this diagram, shapes are used to differentiate precursors while colours differentiate cities. To reflect the

differences in potentials (concentration change resulting from an emission reduction) of different precursors, the size of a symbol is set proportional to the maximum potential found over all precursors and all models, for each city. Finally, we use symbol filling to distinguish cases where the modelled responses change signs (filled symbol) between models (i.e. a positive vs a negative concentration change).

We also use the median concept as discussed in Thunis et al. (2023). The median is calculated from three emission inventories: EDGAR, CAMS221 and EMEP-GNFR. The proposed approach then consists in comparing each model (i.e. EMEP with one inventory) with the median to identify inconsistencies (see Thunis et al. 2023 for more details). The median is not meant here to represent a more accurate model response but rather as a common benchmark to compare models to.


### 3. Results

In this section we first assess the level of consistency in terms of input emissions, the driving factor for potential differences, before analysing their impact in terms of concentration changes.

#### 3.1 Analysis of the emissions

Analysing the PPM emissions from the four emission inventories in Fig. 1, we see that all emissions compare well in general, apart from EMEP-GNFR in Bucharest (lower) and CAMS221 in Stockholm (lower). EDGAR (red coloured) registers the highest PPM emissions for Malopolska, the Po valley, Rome and Stockholm. The differences in PPM emissions between CAMS221 and CAMS42C can be explained by the replacement in the CAMS221 (REF1) inventory of country reported PM2.5 and PM10 emissions for residential heating by emissions that account for condensables in CAMS42C. Condensables are emitted as gaseous compounds, that immediately condense to form organic aerosols. They lead to overall higher PM emissions in CAMS42C. Significant changes in PM2.5 emission quantities due to the presence of condensables are found for several countries, like Spain, Italy and Romania, while differences are smaller for Germany and France (Kuenen et al., 2022). This corroborates the higher PPM emissions in CAMS42C than CAMS221 for the Po valley, Rome, Madrid and Stockholm found in this study. The emission quantities for each location, pollutant and inventory are given in Table S3 of the Electronic Supplement of this study.

Furthermore, Kuenen et al., (2022) showed that the emission differences between CAMS221 and CAMS42C can be explained by the different methodologies and recalculation of the officially reported emissions. Also, each year, an update is processed of the past country's reported emissions based on latest information of activity data or emission factors (EFs). This helps to explain the differences between the emission inventories and reported years.

Kuenen et al., (2022) also showed that in general, for Europe, NMVOC, NH3, PPM10, PPM2.5, NOx and SO2 emissions are higher in EDGAR than in CAMS42, with larger differences for non-EU countries. This could be explained by the fact that EDGAR uses a bottom-up methodology instead of the reported country totals, which has been shown to have, in general, higher uncertainties (Cheewaphongphan et al., 2019).

In our study, we compare the emission densities for smaller areas, but we find similar differences, i.e. EDGAR registers higher emissions for the above-mentioned pollutants for the eight areas considered in our study, apart from yearly NOx emissions for Bucharest, Madrid, Malopolska, Rome and Po Valley, where the emission densities are similar to the other emission inventories. Also, there are substantial differences in PPM emissions for Bucharest between EMEP-GNFR (3.1 mg/m2/day) and the other three inventories, 7.6 mg/m2/day for EDGAR, 8.0 mg/m2/day for CAMS221 and 8.2 mg/m2/day for CAMS42C, which clearly impact the model responses (in terms of concentration) to emission reductions, as will be further discussed in the next section.

Overall, SOx, NH3 and VOC by EMEP-GNFR and the two CAMS inventories agree well, while EDGAR generally shows higher emission densities for these pollutants except for Bucharest and Po Valley for VOC and NH3 for Madrid and Bucharest.

More details on the explanation regarding the differences between CAMS221, CAMS42C and EDGAR are described in Kuenen et al., (2022). At the urban scale, Thunis et al. (2021b) showed that for some sectors and pollutants, the EDGAR emissions were significantly larger than other inventories. This is the case for the SO2 emissions from the industrial sector because of differences in terms of country totals but also in terms of spatial proxies used.

### 3.2 Variability of PM10 BaseCase concentrations

Yearly averaged PM10 concentrations from EDGAR are in general higher than with the other emission inventories, except for Brussels, Madrid and Stockholm (Fig. 2). We have seen in section 3.1 that for some locations PPM, SOx, NH3 or NMVOC emissions from EDGAR are higher than the other three inventories. However, differences in emissions do not lead to important differences in terms of PM10 concentrations.

For Malopolska we find that PM10 values by CAMS42C are higher than CAMS221 and EMEP-GNFR due to the inclusion of condensables for residential heating. Note that inclusion of condensables leads to larger differences over the eastern part of Europe.

Interesting to mention is the large difference in PM10 concentrations for Bucharest between EMEP-GNFR and the other three inventories (EMEP-GNFR lower), which can be explained, at least partly, by the differences in PPM emissions, as mentioned in Section 3.1.

### 3.3 Analysis of potentials and potencies for PM10

Fig. 3 represents the impact on calculated PM10 concentrations of emission reduction of NH3, VOC, NOx, PPM and SOx for the different locations. The plots show the potency on the y-axis, the emissions on the x-axis and the potential (obtained at 50%) along descending diagonal (indicated with dashed lines). The diamond shape (delineated by black bold lines), indicate that differences in emissions, potencies and potentials between a given model and the median are below 20%, where the median is calculated from three emission inventories: EDGAR, CAMS221 and EMEP-GNFR. The 'fac2' lines indicate a factor of two difference as compared to the median, respectively. The consistency indicator (top right) provides information on the percentage of pollutant/city (p,c) couples that fall within the diamond shape, e.g. in the case of CAMS42C, 50% of the (p,c) couples show differences with the median estimate that remain below 20%. Below we analyse the results per precursor.

### 3.3.1 PPM

EMEP-GNFR calculates much higher potentials for PM10 in Stockholm, due to an overestimation of the PPM potency by a factor ~2, see Fig. 3b. For Bucharest a much lower potential for PM10 is found, which can be explained by the underestimation (around a factor 2) of the PPM emissions. Also, CAMS221 displays lower PPM emissions for Stockholm (Fig. 3c), but these lower emissions are compensated by higher potencies (factor ~2 higher), leading to similar PM10 potentials as for the other inventories. For Berlin, the CAMS221 PPM potency is more than a factor 2 lower, leading to underestimation in terms of potential of a factor ~2, despite a slight overestimation of the emissions. Because PPM does not undergo chemical reactions, we expect a relatively linear relationship between emissions and concentrations. In other words, we expect emissions and potentials to be correlated (e.g. Bucharest for EMEP-GNFR). In some instances, this is however not the case (e.g. Stockholm for EMEP-GNFR and CAMS42C). These differences can partly be explained by the sector allocation of the PPM emissions in the four inventories, as shown in Fig. 4. EDGAR assigns much larger PPM emissions in sector 2 (Industry), while EMEP-GNFR has larger PPM emissions in sector 6 (road transport). This is important as emissions are distributed vertically in a different way depending on the sector. Industrial emissions, mainly emitted by stacks at higher levels travel over larger distances and will have less impact on surface concentrations locally than emissions emitted at ground such as road transport. This explains the much higher potencies in EMEP-GNFR for Stockholm. It also stresses the importance of the sectorial repartition of the emissions, especially for PPM that generally shows the largest potencies among all precursors. Another reason for these differences is the spatial distribution of the emissions which differ from one inventory to the other (see supplementary material Fig. S1).

As mentioned before, CAMS42C includes condensables leading to larger PM10 potentials than CAMS221. Despite the overall increase of PPM emissions caused by the inclusion of condensables in CAMS42C, emissions remain lower than the median in cities like Stockholm (red circle in Fig. 3d). For Malopolska the potential is larger for EDGAR and CAMS42C (see also Fig. S2 ES), partly caused by larger emissions.
Apart from the vertical and spatial distribution of the emissions, another reason for differences in potentials might be related to the fact that the location of the P95 (95 Percentile value) values cells where the concentration changes are calculated differ for each model, as shown in Fig.5. More specifically, the P95 values might be positioned at different locations in the 4 Base Cases (see Fig. 5 shaded grid cells).

PM10 includes not only primary particles, but also secondary particles. Secondary particles are formed by gases reacting (such as NOx, VOC and SOx) and condensing (gas to particle conversion) onto pre-existing particles or by nucleation. In the next section we analyse the impact of aerosol secondary precursors reductions on calculated PM10 concentrations.

### 3.3.2 NOx

Compared to other precursors, NOx shows a good agreement among models with a couple of inconsistencies identified in the Po Valley for EDGAR and EMEP42C, where the potencies are slightly larger than the 20% threshold around the median. This good agreement can be explained by the fact that NOx emissions originate in

great part from the transport sector, a sector for which the spatial proxies (for the spatial and sectorial disaggregation) are generally well described and harmonised among inventories (Trombetti et al. 2018). In addition, NOx sources are mostly diffuse (as opposed to point sources) and less subject to localised hot-spot differences.


### 3.3.3    SOx

For Stockholm large differences are found in EDGAR potentials when compared to the median (Fig.3a, indicated by red coloured rectangular box). The explanation for this is a strong overestimation of the SOx emissions (factor ~10, Fig. 1c), which is partly compensated by an underestimation of the potency (factor ~2). For MAD and BRU,

we see that higher SOx emissions (factor ~2) by EDGAR are compensated by lower potencies, which lead to overall similar potentials.

Hence, reducing SOx emissions in EDGAR has a larger impact on PM10 concentrations when compared to the median, via the chemical reactions that lead to the formation of ammonium sulphate aerosol as described in De Meij et al. (2009c).


### 3.3.4    NH3

With the exception of EMEP-GNFR, all models show an inconsistency for NH3 in the Malopolska region. EDGAR shows higher emissions (factor ~2) than the ensemble, but these higher emissions are compensated by lower potencies, which lead to overall similar potentials. CAMS221 and CAMS42C both show larger emissions

too (although to a lesser extent than EDGAR) and lower potencies, leading to relatively similar potentials (green diamond symbols in Fig. 3). Note that given the reduced NH3 emissions in urban areas, these emissions do not lead to important potentials in many cities, hence they do not appear in the diagrams.

### 3.3.5    VOC

VOC potentials are generally too low (lower than the 20% threshold detailed in Section 2.4) to appear in the figures, apart from the Po Valley where CAMS221 shows a small inconsistency with respect to the median (orange squares in Fig. 3).

From the analysis of these different precursors, PPM appears to be the precursor leading to the major differences

in terms of potentials, i.e. in terms of concentration change responses that are of direct relevance when designing air quality plans (Fig. S5). Although simpler to manage because of their linearity, they deserve more attention given their important variability (among models) and importance in terms of final concentrations.

 ### 3.3.6    **SOx/NOx ratios**

To understand better the sensitivity of PM10 formation to NOx, SOx, or NH3 reductions, we analyse the ratios between these precursors across inventories.

Table 2a shows the ratios between BaseCase domain averaged SOx and NOx emission densities. For example, the minimum ratio is around 0.06, indicating that there are around 16 times more NOx (11.5 mg/m2/day) than

SOx emissions (0.74 mg/m2/day) in Rome for EMEP-GNFR. Table 2b shows on the other hand that the corresponding potency ratios are inverted, with much larger efficiencies when reducing SOx than NOx emissions. The same is true in most cities. This can be explained by the fact that NOx has to compete with NH3 to form PM. Whereas SOx emission reductions directly lead to PM changes. While this behaviour is quite general, there is a large variability in its magnitude. In some cities like Brussels, negative ratios appear caused by concentration

increases when NOx emissions are reduced. This corroborates the findings by Clappier et al., (2021) who found that reducing SO2 emissions where abundant is always efficient and relatively linear, as shown also in the next section on non-linearities.

A similar analysis can be performed with NOx to NH3 ratios, see Table S4 of the ES. NOx to NH3 contribute to the formation of ammonium nitrate aerosol, via the reactions NO2 + OH -> HNO3, that reacts (when there is sufficient ammonia available to neutralize all sulphate) with NH3 to form NH4NO3 aerosol, a fraction of PM10. Details on the chemical pathways can be found in Thunis et al., (2021a). As an example, the emission ratio for Rome by EDGAR is 3.3, while the corresponding numbers are 4.9, 5.4 and 4.8 for EMEPC, CAMS42C and

EMEP-GNFR, respectively. While NOx emissions in the four inventories are similar, EDGAR contains almost a factor 2 more NH3 emissions. This means that NH3 is relatively more abundant in EDGAR and its reduction has therefore less impact on concentration. This results in the formation processes being more 'NOx-sensitive' in Rome. Thus, reducing NOx in EDGAR leads to larger impact on PM10 concentrations.


### 3.3.7    **Non-linearities**

Non-linearity in PM responses to emission changes often results from changes in chemical regimes where the formation process is limited by a different species.

Analysing the absolute potentials ratio (50% vs. 25%) in Tables 3 to 7 provides information on the (non-)linearity

of the relationship between emission and concentration changes. If the ratio is close to 1.00, then there is linear correlation between the two. Departure from 1 indicates non-linearity. We only show the ratios which are 3% or higher when compared to the 50% potential for PPM in order to highlight the most relevant ratios.

For primary PM (PM2.5 and PMcoarse) we get a linear relationship as expected, see Table 6. The reason for this is that primary emissions only affect the primary part of the aerosol formation and do not undergo chemical

reactions.

For NOx (Table 3) the behaviour is generally non-linear with ratios larger than 1.00. This indicates that calculated PM10 concentrations would be reduced more between 25 and 50% than between 0 and 25%. For example, EDGAR indicates 1.18 in Rome, indicating that PM10 concentrations would be reduced by 18% more between

25 and 50% than between 0 and 25%. This might be explained by a change of chemical regime from a NH3-limited regime (when NOx is more abundant and less efficient) to a NOx-limited regime (NOx is less abundant and more efficient) as emissions are reduced further.

Note the importance of averaging processes on the indicator value. Based on 95-percentile locations, the ratio for Bucharest with EMEP-GNFR is 1.18 whereas for domain-averaged values the ratio becomes 1.08 indicating closer to linear relationships. This corroborates the results by Thunis et al., (2021c), who assessed the contribution of cities to their own air pollution. They showed that the type of indicator impacts the final outcome, i.e. the share of the city pollution caused by their own emissions in his study. It also confirms that indicators based on averaged values tend to report more linear relationships.

For VOC (Table 4) and SOx (not shown, as the ratios compared to PPM are less than 3%) we find that ratios remain very close to 1.00.
NH3 shows significant non-linearity with ratios larger than 1 (Table 5). The same explanations as for NOx can be used to explain the larger efficiency of emission reductions when these emissions are reduced further in a NH3-limited regime.

Finally, a similar ratio can be constructed for emission reductions that include all species together (SOx, NOx, VOC, NH3, PM2.5 and PMcoarse). The results generally indicate a linear behaviour mainly because of compensating effects (NOx non-linearities are weakened by other emitted species), with the exception of EMEP-GNFR in Berlin and BUC. For these two locations, the explanation lies in the much lower PPM emissions (linear) and larger NOx emissions (non-linear). Clappier et al. (2021) showed which chemical regimes are responsible to the secondary inorganic PM formation over Europe, and how these chemical regimes can help in designing efficient PM abatement strategies. They showed that during wintertime, PM25 concentrations are predominantly NH3-sensitive in the major part Europe. During summertime, PM25 are predominantly SO2-sensitive in most of Europe.

### 3.4 Variability of ozone BaseCase concentrations

Ozone is chemically formed by the oxidation process of volatile organic compounds (VOCs) in the presence of NOx (NO + NO2) and its formation is driven by the sunlight intensity. At the same time, NOx also works as an ozone sink through NOx titration (NO+O3→NO2+O2) that occurs during the night and wintertime, i.e. less photolysis reactions of NO2 (Jhun et al., 2015) and O3 is removed by NO emissions from road traffic in city centres (Sharma et al.,2016).
Fig. 6 shows that yearly averaged O3 concentrations are very similar.

### 3.5 Analysis of potentials and potencies for O3

In Fig. 7 we analyse the impact of the reduction of NOx and VOC on calculated O3 concentrations for the different locations.

The production of O3 depends on the availability of NOx and VOCs, which are emitted mostly from sectors such as industry and road transport. For that reason, only NOx and VOC appear in Fig. 7, except for NH3 for EDGAR. The latter can be explained by the fact that NH3 contributes to the formation of secondary aerosol and decreases the acidity of the aerosols. The aerosol pH plays an important role in the reactive uptake and release of gases, which can affect ozone chemistry (Pozzer et al., 2017). This NH3 impact also exists for the other inventories but is lower than the 20% threshold and therefore does not appear in the diamond plots.

### 3.5.1    VOC

With the exception of CAMS221, all models show some differences with the median for VOC (Fig. 7). In Maloposka, Stockholm and Berlin, EDGAR emissions are a factor ~2 higher than the median value. While in the first location, the lower potencies compensate for these emission differences, leading to similar potentials. This is not the case for the two latter cities, where similar potencies lead to larger potential. For EMEP-GNFR, only Bucharest shows differences with lower emissions and higher potencies, leading to similar potentials. It is interesting to note the large differences between CAMS221 and CAMS42C. While the addition of condensable in CAMS42C does not impact O3 formation, other changes included in version CAMS42C have significant impacts. While NOx responses dominate in most cases in CAMS221, this is not the case in CAMS42C where VOC responses become important for three cities. Differences with the median are mostly caused by potency rather than by emission differences. This is an interesting information that a change of version can lead to very important changes in model responses despite similar absolute O3 levels.

Note that VOC appears systematically as an important impact (i.e. visible in the diagram) for Malopolska and Po Valley, whereas this is not the case systematically for the other locations. The reason is that for these two regions, emissions are reduced over larger areas, leading to larger impacts. More details on the potentials and potencies for the different locations can be found in the supplement material Fig. S6 – S8.

### 3.5.2    NOx

NOx shows generally larger impacts than VOC (see supplementary material Fig. S9). While for PM10, NOx responses were shown in the previous section to be consistent among models; this is not the case for O3. Potential differences originate mostly from differences in potencies while emissions remain relatively similar among inventories. The largest differences occur for Bucharest (EMEP-GNFR), Malopolska and the Po valley (CAMS42C) with much larger potency estimates than the median, indicating that these regions are more sensitive to NOx reduction than for other inventories. However, opposite trends also occur as in Berlin for CAMS221 and CAMS42C. It is also interesting to note that in some cities like Brussels, differences in model versions (CAMS42C vs. CAMS221) significantly affects the NOx responses (as already noted for VOC).

In Malopolska, EDGAR and CAMS42C show a change of sign in terms of responses. In such cases, NOx reductions lead to an O3 increase whereas the median shows an opposite behaviour.

The highest consistency (84%) with the ensemble is found for EMEP-GNFR, meaning that 84% of the relevant impacts (delta concentrations) are within the 20% limit, indicating that EMEP-GNFR is often picked as the median. On the other hand, CAMS42C and EDGAR show the lowest consistency value. It is interesting to note the large difference between the two versions of the same inventory (60 vs. 35% for CAMS221 and CAMS42C, respectively).


Similarly, to PM10, some of the differences are partly explained by the location of the P95 values that are not similar for the four inventories, as shown in Fig.8, where EDGAR locations differ from all others (shaded grid cells).

**3.5.3    VOC/NOx ratios**

To understand better the impact of NOx and VOC reductions on the production or loss of O3, and the interconnections between the two, we analyse the VOC/NOx ratio for the different inventories in Table 8a. For Malopolska, Bucharest or Brussels, the VOC/NOx emission ratio for EDGAR is twice as large than the others. This reflects in the EMEP-GNFR diagram where these cities show clear inconsistencies. The larger amount of

VOC in these cities does not impact significantly the potencies (Table 8b). While NOx potencies are mostly positive, indicating an increase of the O3 concentrations over the urban areas, VOC potencies are always negative, indicating lower O3 concentrations when reducing VOC emissions. Differences in VOC/NOx ratios might lead to changes of chemical regime, that explain some of the differences in the potentials.

The differences in VOC/NOx ratios between the four emission inventories highlight the importance of the accuracy of emission inventories, which could strongly impact the chemical regime (i.e. NOx-limited or VOC-limited). Even moderate perturbations in NOx or VOC emissions could change the chemical regime of O3 formation (Xiao et al. 2010).


**3.5.4    Non-linearities**

Previous studies (Cohan et al. 2005, Xiao et al., 2010) have shown that the formation of ozone is more sensitive to large reductions of NOx that depart from a linear emission scaling. To this end, we show in Table 9 and 10 the ratio between absolute potentials (at 50% and 25%) for P95, which help to assess the level of non-linearity of the

atmospheric reactions that involve gaseous precursors NOx, and VOCs in the formation of ozone. Table 9 shows large non-linearities when NOx emissions are reduced. A number larger than 1 indicates superlinearity; that means that O3 concentrations are more reduced between 25 and 50% than between 0 and 25%. For Malopolska, we find a large ratio for EDGAR (12.03) because it is based on small values (-0.325 vs. -0.027).

Ratios are generally lower than one, with the clear exception in the Po Valley. This must be put in relation with

the fact that the Po valley is the only place where potencies are negative (see Table 8b), indicating a different chemical regime (O3 formation) than in other locations (O3 titration). This is explained by the fact that the Po Valley domain includes sub-urban and background areas where O3 formation takes place.

For VOC the ratios are close to 1.00 indicating a linear behaviour (Table 10). This corroborates previous studies (Xiao et al., 2010).

Reducing NOx and VOC emissions together (Table 11) also shows some non-linear behaviour that originates from the NOx side. The formation of O3 is less sensitive to the reduction of NOx emissions when VOC emissions are simultaneously reduced. This corroborates the findings of Xiao et al., 2010, Xing et al., 2017.

**4.       Concluding remarks**


In this work, we assessed how emissions impact the model the BaseCase concentrations but also concentration changes when emission reductions are applied. The impact of emission reductions based on four emission inventories (EDGAR 5.0, EMEP-GNFR, CAMS version 2.2.1 and CAMS version 4.2 + condensables) has been investigated for PM10 and O3 in eight cities/regions in Europe. We assessed the model's variability in terms of

model responses to emission changes with the support of specific indicators (potentials and potencies) and used a screening method adapted from Thunis et al. (2022) to identify the main inconsistencies among model responses. A median value has been constructed to serve as reference for the comparisons.

Our study reveals that the impact of reducing aerosol (precursors), such as PPM, NOx, SO2, NH3, VOCs result in different potentials and potencies, differences that are mainly explained by differences in emission quantities, differences in their spatial distributions as well as in their sector allocation. The main findings are the following:

- In general, the variability among models is larger for concentration changes (potentials) than for absolute
concentrations. This is true for both PM10 and O3.
- Emission densities at each location for all precursors are quite consistent apart from EDGAR which generally show larger urban scale emissions.
- Similar emissions can however hide large variations in sectorial allocation. Our results stress the importance of the sectorial repartition of the emissions, given their different vertical distribution
(emissions in the industrial sector are emitted at higher levels and have less impact on surface concentration) especially for PPM. This sectorial allocation can lead to large impacts on potency. For similar reasons, larger emissions do not necessarily lead to larger potencies. At the local scale, it is therefore important to further work on the modelling of PPM and on the estimate of its underlying emissions.
- PPM appears to be the precursor leading to the major differences in terms of potentials, i.e. in terms of PM10 concentration changes. This is of direct relevance when designing air quality plans. Although simpler to manage because of their linearity, they would deserve more attention at the local scale given their importance in terms of final concentrations and their large variability (among models). Additional efforts to check the consistency and accuracy of the PPM emissions and their sectorial share is therefore
important to ensure robust model responses.
- For O3, NOx emission reductions are the most efficient, likely because of the urban focus of this work and the abundance of NOx emission in this type of areas.

- In terms of non-linear behavior, the relationship between emission reduction and PM10 concentration change shows the largest non-linearity for NOx and in a lesser extend for NH3 whereas it remains mostly linear for the other precursors (VOC, SOx and PPM). Potentials based on a single emission reduction value are therefore most of the time not sufficient and do not provide a complete view of the non-linear behaviour of the emission reductions. Additional NOx emission reductions are necessary to better understand the non-linearity of reducing VOC and NOx reductions together.

- In terms of non-linear behaviour, the relationship between emission reduction and O3 concentration change shows the largest non-linearity for NOx (concentration increase) and a quasi linear behaviour for VOC (concentration decrease). Similar to above, potentials based on a single emission reduction value for NOx are not always sufficient to understand the non-linear behaviour of emission reductions.

- Potencies and potentials can show differences that are as large between inventories (CAMS221 vs EMEP-GNFR) than between inventory versions (CAMS221 vs. CAMS42C). This is the case for example in Brussels for the NOx responses on PM10 concentrations.

- Precursor emission ratios (e.g. VOC/NOx for ozone or NOx/NH3 for PM10) show important differences among emission inventories. This emphasizes the importance of the accuracy of emission estimates since these differences can lead to changes of chemical regimes, directly affecting the responses of O3 or PM10 concentrations to emission reductions.

- It is also important to understand that the choice of the indicator used in a given analysis (for example mean or percentile values) can lead to different outcomes. It is therefore important to assess the variability of the results around the choice of the indicator to avoid misleading interpretations of the results.

From an emission inventory viewpoint, this work indicates that the most efficient actions to improve the robustness of the modelling responses to emission changes would be to better assess the sectorial share and total quantities of PPM emissions. Another important aspect is to better assess emitted precursor ratios as these lead to important differences in term of model responses, both in the case of O3 (NOx/VOC ratio) and PM (NOx/NH3/SOx ratios). From a modelling point of view, NOx responses are the more challenging and require caution because of their non-linearity.

**Code availability**

The source code of the screening method of the statistical analysis can be found here:
https://doi.org/10.5281/zenodo.8082531

**Data availability**

The emission data can be downloaded here: https://eccad.sedoo.fr/

**Author contributions**

AdM and PT wrote the manuscript draft; AdM, PT and CC produced the data; AdM, PT and CC analyzed the data; EP and BB reviewed and edited the manuscript.

**Competing interests**

The authors declare that they have no conflict of interest.

**Acknowledgment ECCAD**

The authors would like to thank Emissions of atmospheric Compounds and Compilation of Ancillary Data
(ECCAD) system (https://eccad.aeris-data.fr) for archiving and distribution of the emission inventories.

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

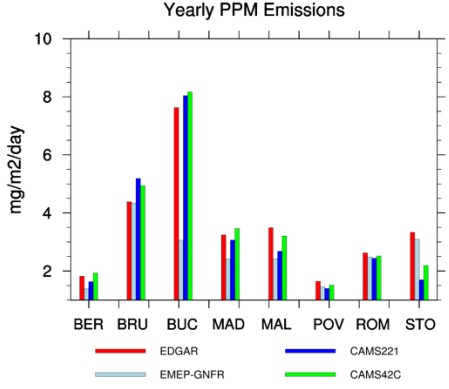

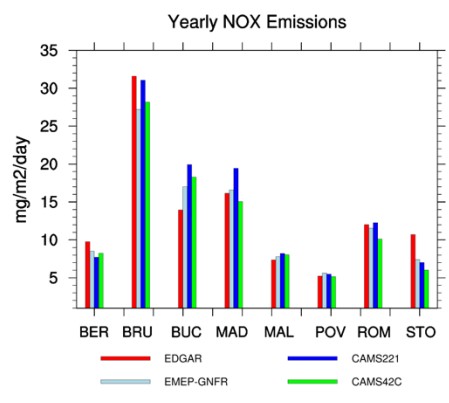

(a)  (b)

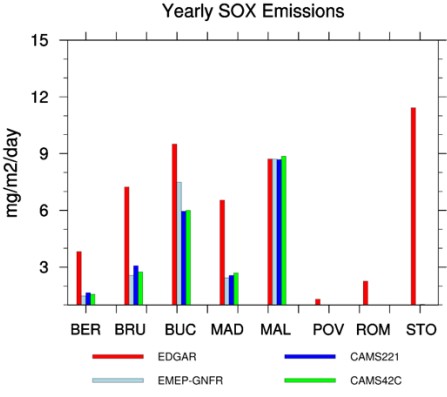

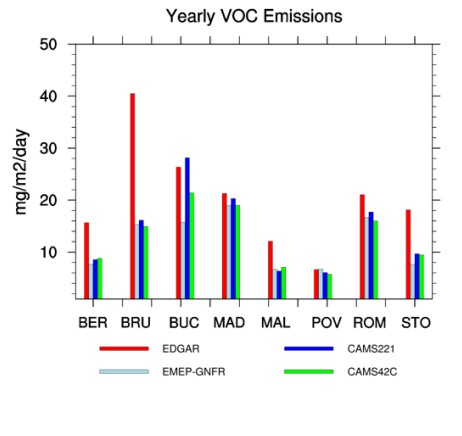

(c)  (d)

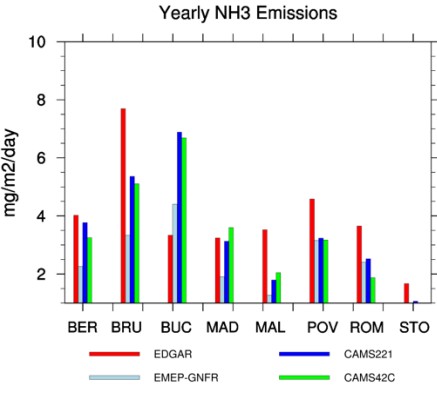

(e)

**Figure 1. Annual mean emission densities (mg/m2/day) for (a) PPM, (b) NOx, (c) SOx, (d) VOC and (e) NH3 by EDGAR (red), EMEP-GNFR (light blue), CAMS221 (blue) and CAMS42C (green), for the eight locations (Berlin [BER], Brussels [BRU], Bucharest [BUC], Madrid [MAD], Malopolska region [MAL], Po Valley region [POV], Rome [ROM] and Stockholm [STO]).**


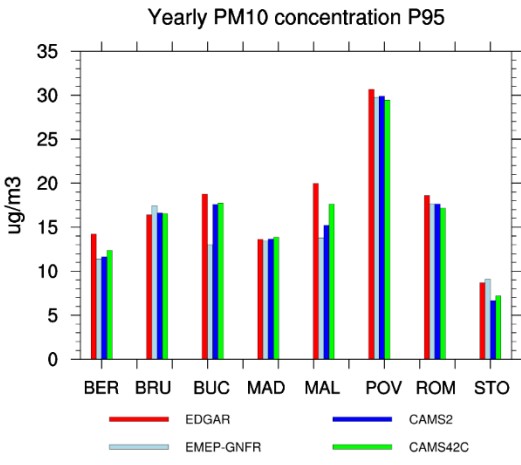

**Figure 2. Total yearly average PM10 concentrations by EDGAR (red), EMEP-GNFR (light blue), CAMS221 (blue) and CAMS42C (green), for the eight locations (Berlin [BER], Brussels [BRU], Bucharest [BUC], Madrid [MAD], Malopolska region [MAL], Po Valley region [POV], Rome [ROM] and Stockholm [STO]). The concentrations represent values above the 95 Percentile values, showing the highest 5% values in the domain from the BaseCase.**

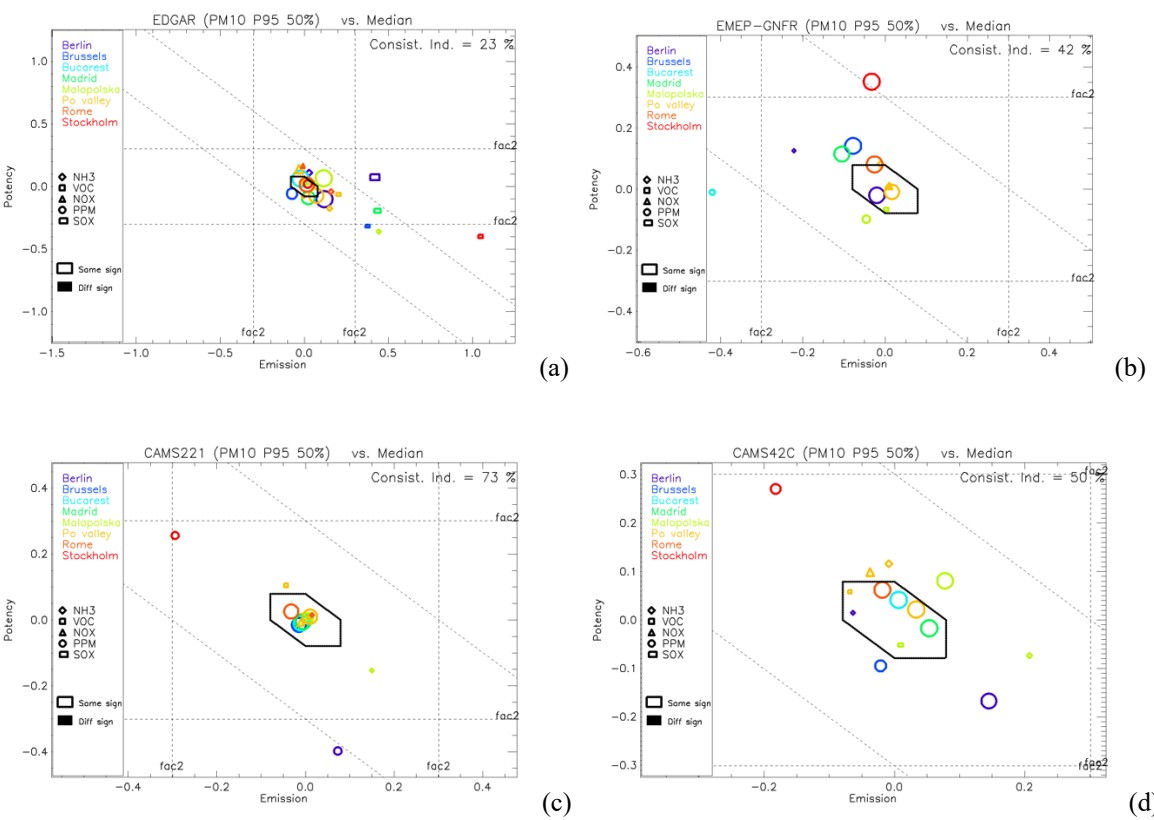

**Figure 3. Diamond plot for PM10 concentrations for (a) EDGAR, (b) EMEP-GNFR, (c) CAMS221 and (d) CAMS42C. The values represent values above the 95 Percentile, showing the highest 5% values in the domain from the BaseCase. The X- and Y-axis are expressed as logarithms. For each city, the size of a symbol is proportional to the maximum absolute potential of the considered precursor, across models. Note that symbols for which emissions are relevant and that characterise the median all fall at the (0,0) position. For visualisation purpose, these have been slightly shifted within the diamond shape.**

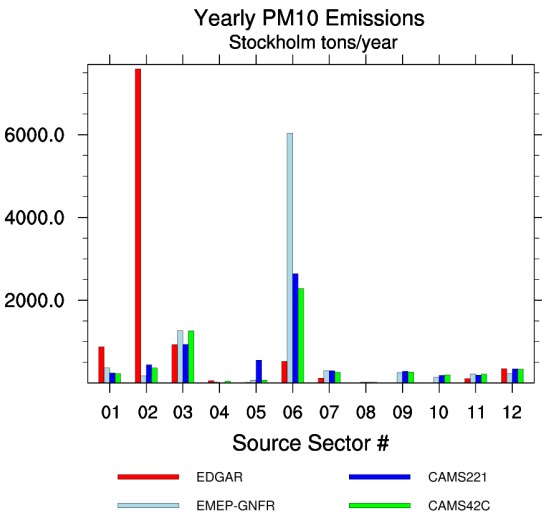

**Figure 4. Total PM10 emissions (tons/year) for Stockholm for EDGAR (red), EMEP-GNFR (light blue), CAMS221 (blue) and CAMS42C (green) for each GNFR sector.**


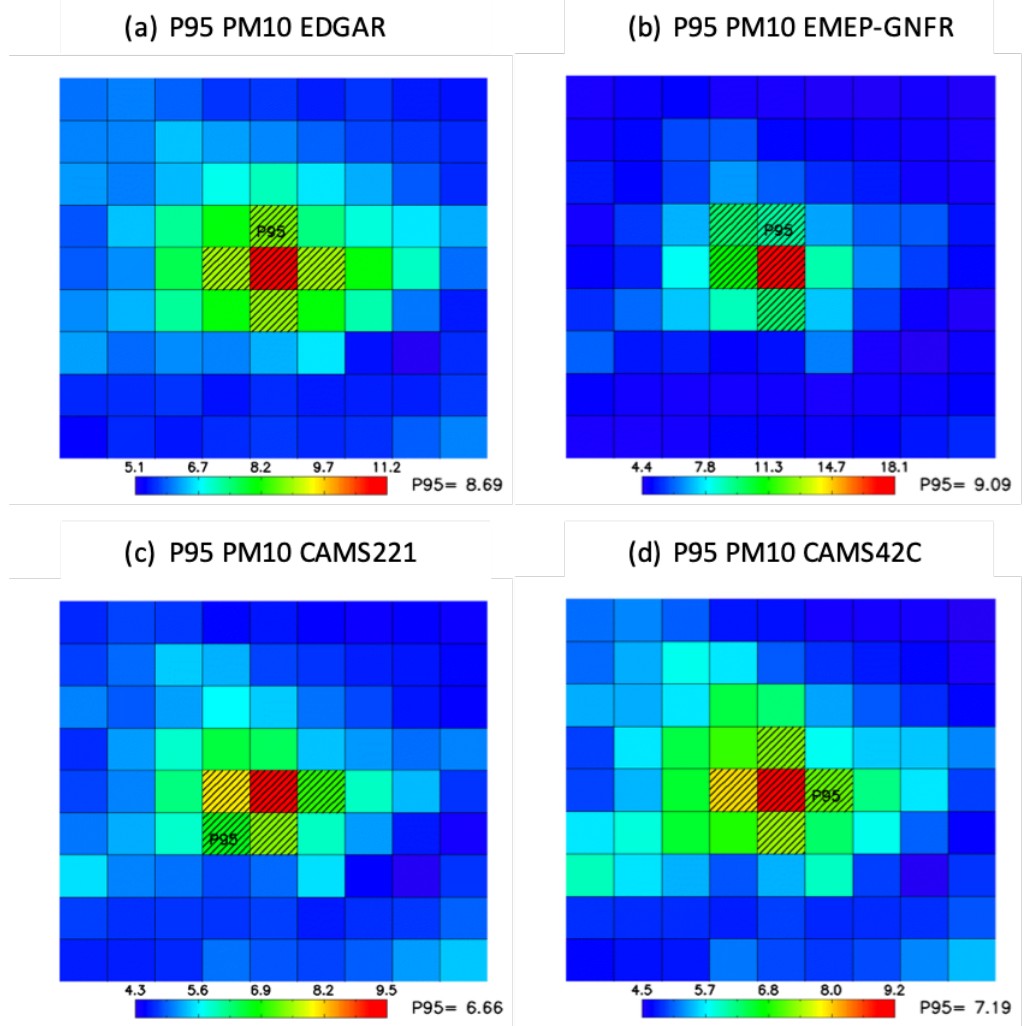

**Figure 5. Overview of the location of the P95 values for the calculated PM10 concentrations µg/m3 by the four Base Cases for the domain STO. Shaded grid cells indicate the location of the values above the P95 by (a) EDGAR, (b) EMEP-GNFR, (c) CAMS221 and (d) CAMS42C. The number next to P95 represents the average of the P95 values.**

Table 1. Overview of the four emission inventories used in this study.

| Name inventory | Resolution (lon x lat) in degrees | Method | Release date | Sector classification | Condensables included | Total NOx emissions* | Total SOx emissions* | Total PM25 emissions* | Total NH3 emissions* |
|---|---|---|---|---|---|---|---|---|---|
| Edgar_v5.0 | 0.1 x 0.1 | Bottom-up | 2020 | 13 GNFR | No | 6360 | 4074 | 1278 | 5116 |
| EMEP | 0.1 x 0.1 | Country report | 2018 | 13 GNFR | No | 7445 | 2591 | 1229 | 3663 |
| CAMS 2.2.1 | 0.1 x 0.05 | Country report | 2018 | 13 GNFR | No | 6410 | 2513 | 1272 | 3708 |
| CAMS4.2C | 0.1 x 0.05 | Country report | 2022 | 12 GNFR | Yes | 6419 | 2519 | 1688 | 3640 |

*Total emissions for Austria, Belgium, Bulgaria, Denmark, Finland, France, Greece, Hungary, Ireland, Italy, Luxembourg, Netherlands, Poland, Portugal, Romania, Spain, Sweden, Estonia, Latvia, Lithuania, Czech Republic, Slowakia, Slovenia ,Croatia, Cyprus, Malta and Germany in Ktons/year.

**Table 2(a) Overview of BaseCase emissions (mg/m2/day) for NOx and SOx, together with the ratio in the emissions between these two pollutants. (b) Similar as to (a) but for potency at P95 in µg/m3.**

**(a)**

| Emissions mg/m2/day | | | | | Ratio emissions SOx/NOx | | | |
| --- | --- | --- | --- | --- | --- | --- | --- | --- |
| NO$_x$ | EDGAR | EMEP-GNFR | CAMS221 | CAMS42C | EDGAR | EMEP-GNFR | CAMS221 | CAMS42C |
| BER | 9.76 | 8.50 | 7.72 | 8.24 | 0.39 | 0.17 | 0.21 | 0.19 |
| BRU3 | 31.57 | 27.21 | 31.05 | 28.16 | 0.23 | 0.09 | 0.10 | 0.10 |
| BUC | 13.94 | 17.01 | 19.93 | 18.28 | 0.68 | 0.44 | 0.30 | 0.33 |
| MAD | 16.15 | 16.59 | 19.44 | 15.05 | 0.40 | 0.15 | 0.13 | 0.18 |
| MAL | 7.36 | 7.79 | 8.20 | 8.04 | 1.18 | 1.12 | 1.06 | 1.10 |
| POV | 5.23 | 5.62 | 5.48 | 5.16 | 0.25 | 0.13 | 0.15 | 0.16 |
| ROM | 12.01 | 11.54 | 12.26 | 10.11 | 0.19 | 0.06 | 0.08 | 0.08 |
| STO | 10.71 | 7.41 | 7.01 | 6.03 | 1.07 | 0.09 | 0.15 | 0.13 |

| Emissions mg/m2/day | | | | |
| --- | --- | --- | --- | --- |
| SO$_x$ | EDGAR | EMEP-GNFR | CAMS221 | CAMS42C |
| BER | 3.82 | 1.48 | 1.65 | 1.57 |
| BRU3 | 7.23 | 2.56 | 3.07 | 2.75 |
| BUC | 9.51 | 7.49 | 5.95 | 6.00 |
| MAD | 6.54 | 2.43 | 2.56 | 2.69 |
| MAL | 8.72 | 8.71 | 8.69 | 8.86 |
| POV | 1.31 | 0.73 | 0.82 | 0.81 |
| ROM | 2.25 | 0.74 | 0.94 | 0.79 |
| STO | 11.43 | 0.67 | 1.03 | 0.80 |

**(b)**

| Potency P95 (µg/m$^3$/ton) | | | | | Ratio Potency SOx/NOx | | | |
| --- | --- | --- | --- | --- | --- | --- | --- | --- |
| NO$_x$ | EDGAR | EMEP-GNFR | CAMS221 | CAMS42C | EDGAR | EMEP-GNFR | CAMS221 | CAMS42C |
| BER | -0.0018 | -0.0011 | -0.0024 | -0.0018 | 12.4 | 17.1 | 4.9 | 9.9 |
| BRU3 | 0.0013 | 0.0013 | 0.0015 | 0.0012 | -33.3 | -36.2 | -60.1 | -42.7 |
| BUC | -0.0067 | -0.0047 | -0.0019 | -0.0017 | 7.1 | 12.8 | 36.2 | 43.5 |
| MAD | 0.0002 | -0.0002 | -0.0002 | -0.0013 | -179.0 | 280.0 | 223.5 | 27.4 |
| MAL | -0.001 | -0.0011 | -0.0004 | -0.001 | 2.4 | 1.7 | 5.5 | 1.9 |
| POV | -0.0064 | -0.0047 | -0.0046 | -0.0059 | 1.6 | 2.4 | 2.5 | 1.2 |
| ROM | -0.022 | -0.0076 | -0.0151 | -0.0089 | 2.7 | 15.9 | 4.6 | 13.5 |
| STO | -0.0011 | -0.0011 | -0.0006 | -0.0005 | 18.8 | 30.5 | 86.3 | 81.2 |

| Potency P95 (µg/m$^3$/ton) | | | | |
| --- | --- | --- | --- | --- |
| SO$_x$ | EDGAR | EMEP-GNFR | CAMS221 | CAMS42C |
| BER | -0.0223 | -0.0188 | -0.0117 | -0.0178 |
| BRU3 | -0.0433 | -0.0471 | -0.0902 | -0.0512 |
| BUC | -0.0476 | -0.06 | -0.0687 | -0.0739 |
| MAD | -0.0358 | -0.056 | -0.0447 | -0.0356 |
| MAL | -0.0024 | -0.0019 | -0.0022 | -0.0019 |
| POV | -0.01 | -0.0113 | -0.0115 | -0.0072 |
| ROM | -0.0584 | -0.1209 | -0.0695 | -0.1204 |
| STO | -0.0207 | -0.0335 | -0.0518 | -0.0406 |

**Table 3. Absolute potential (50%) divided by the Absolute potential (25%) for PM10 when NOx emissions are reduced by 50% and 25% for 95 Percentile values (P95). Numbers with a ratio higher than 3% compared to the PPM 50% Potential P95 are shown.**

| City | EDGAR | EMEP-GNFR | CAMS221 | CAMS42C |
|------|-------|-----------|---------|---------|
| BER | 1.17 | 1.19 | 1.09 | 1.14 |
| BRU3 | 1.22 | | 1.21 | 1.15 |
| BUC | | 1.18 | | |
| MAD | | | | |
| MAL | 1.17 | 1.14 | 1.29 | 1.20 |
| POV | 1.21 | 1.27 | 1.42 | 1.24 |
| ROM | 1.18 | 1.15 | 1.21 | 1.19 |
| STO | | | | |

**Table 4. Absolute potential (50%) divided by the Absolute potential (25%) for PM10 when VOC emissions are reduced by 50% and 25% for 95 Percentile values (P95). Numbers with a ratio higher than 3% compared to the PPM 50% Potential P95 are shown.**

| City | EDGAR | EMEP-GNFR | CAMS221 | CAMS42C |
|------|-------|-----------|---------|---------|
| BER | | | | |
| BRU3 | | | | |
| BUC | | | | |
| MAD | 0.97 | | 0.96 | |
| MAL | | | | |
| POV | | 0.97 | | 0.99 |
| ROM | | | | |
| STO | | | | |

**Table 5. Absolute potential (50%) divided by the Absolute potential (25%) for PM10 when NH3 emissions are reduced by 50% and 25% for 95 Percentile values (P95). Numbers with a ratio higher than 3% compared to the PPM 50% Potential P95 are shown.**

| City | EDGAR | EMEP-GNFR | CAMS221 | CAMS42C |
|------|-------|-----------|---------|---------|
| BER | 1.11 | 1.08 | 1.11 | 1.09 |
| BRU3 | 1.09 | 1.11 | 1.09 | 1.11 |
| BUC | 1.09 | 1.08 | 1.12 | 1.10 |
| MAD | 1.15 | 1.09 | 1.12 | 1.13 |
| MAL | 1.16 | 1.21 | 1.03 | 1.13 |
| POV | 1.28 | 1.28 | 1.26 | 1.26 |
| ROM | 1.15 | 1.13 | 1.16 | 1.14 |
| STO | 1.13 | 1.15 | 1.10 | 1.04 |

**Table 6. Absolute potential (50%) divided by the Absolute potential (25%) for PM10 when PM2.5 and PMcoarse emissions are reduced by 50% and 25% for 95 Percentile values (P95).**

| City | EDGAR | EMEP-GNFR | CAMS221 | CAMS42C |
|------|-------|-----------|---------|---------|
| BER | 1.00 | 1.00 | 1.00 | 1.00 |
| BRU3 | 1.00 | 1.00 | 1.00 | 1.00 |
| BUC | 1.00 | 1.00 | 1.00 | 1.00 |
| MAD | 1.00 | 1.00 | 1.00 | 1.00 |
| MAL | 1.00 | 1.00 | 1.00 | 1.00 |
| POV | 1.01 | 1.01 | 1.01 | 1.01 |
| ROM | 1.00 | 1.00 | 1.00 | 1.00 |
| STO | 1.00 | 1.00 | 1.00 | 1.00 |

**Table 7. Absolute potential (50%) divided by the Absolute potential (25%) for PM10 when ALL pollutants (SOx, NOx, VOC, NH3, PM2.5 and PMcoarse) emissions are reduced together by 50% and 25% for 95 Percentile values (P95). Numbers with more than 5% non-linearity are highlighted.**

| City | EDGAR | EMEP-GNFR | CAMS221 | CAMS42C |
|------|-------|-----------|---------|---------|
| BER | 1.01 | 1.24 | 1.02 | 1.01 |
| BRU3 | 1.00 | 1.11 | 1.01 | 1.01 |
| BUC | 1.01 | 1.19 | 1.01 | 1.00 |
| MAD | 1.01 | 1.02 | 1.01 | 1.01 |
| MAL | 1.01 | 1.07 | 1.01 | 1.01 |
| POV | 1.02 | 1.05 | 1.02 | 1.00 |
| ROM | 1.01 | 1.04 | 1.01 | 1.01 |
| STO | 1.01 | 1.03 | 1.01 | 1.00 |




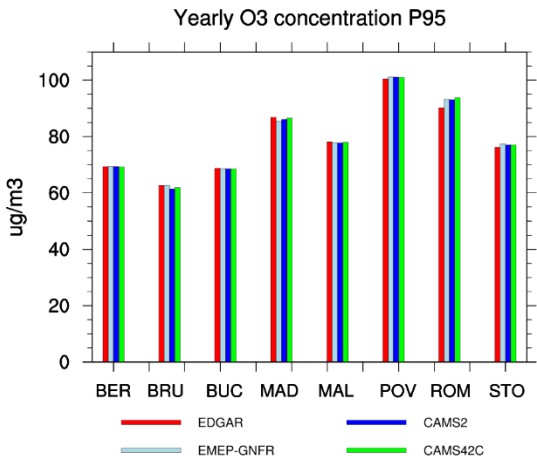

Yearly O3 concentration P95

**Figure 6. Yearly average O3 concentrations by EDGAR (red), EMEP-GNFR (light blue), CAMS221 (blue) and CAMS42C (green), for the eight locations (Berlin, Brussels, Bucharest, Madrid, Malopolska region, Po Valley region, Rome and Stockholm). The concentrations represent values above the 95 Percentile values, showing the highest 5% values in the domain from the BaseCase.**


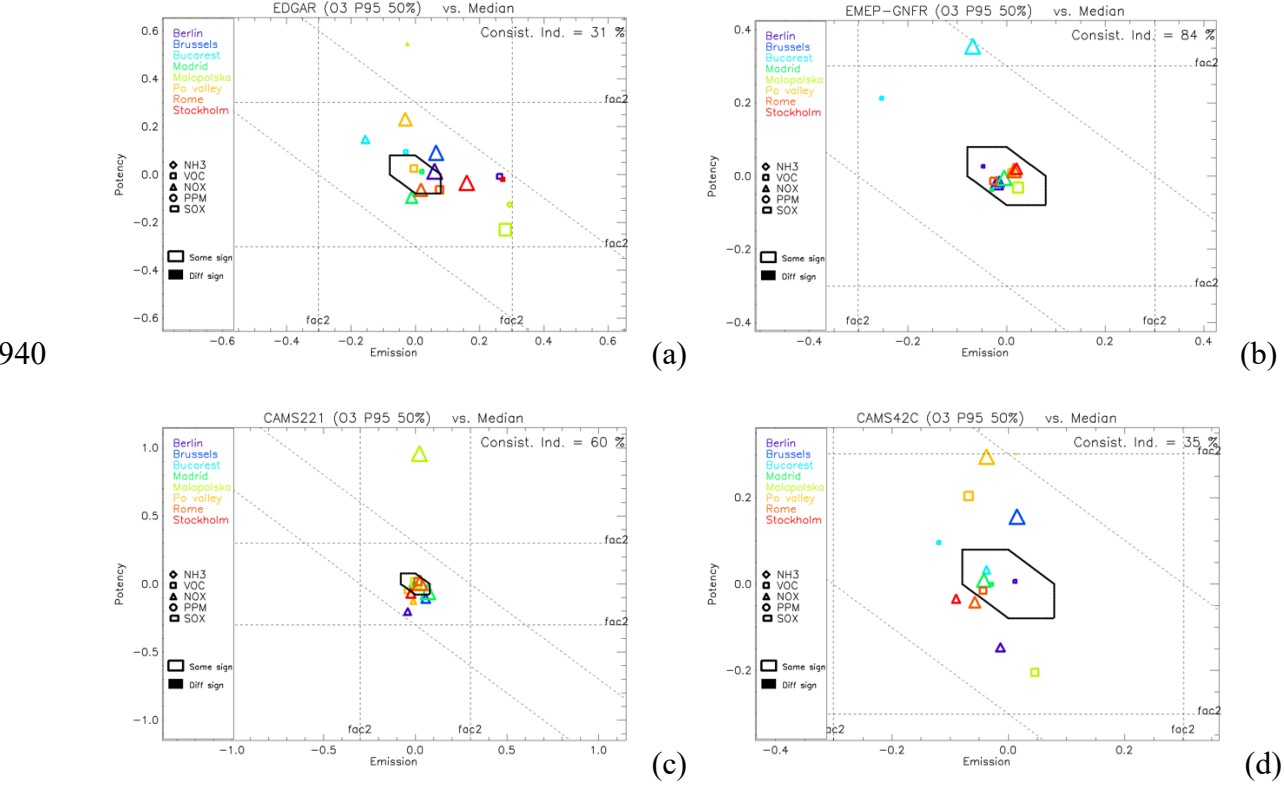


(a)          (b)

(c)          (d)

**Figure 7. Diamond plot for O3 concentrations for (a) EDGAR, (b) EMEP-GNFR, (c) CAMS221 and (d) CAMS42C. The values represent values above the 95 Percentile, showing the highest 5% values in the domain from the BaseCase. The X- and Y-axis are expressed as logarithms. For each city, the size of a symbol is proportional to the maximum absolute potential of the considered precursor, across models. Note that symbols for which emissions are relevant and**


that characterise the median all fall at the (0,0) position. For visualisation purpose, these have been slightly shifted within the diamond shape.


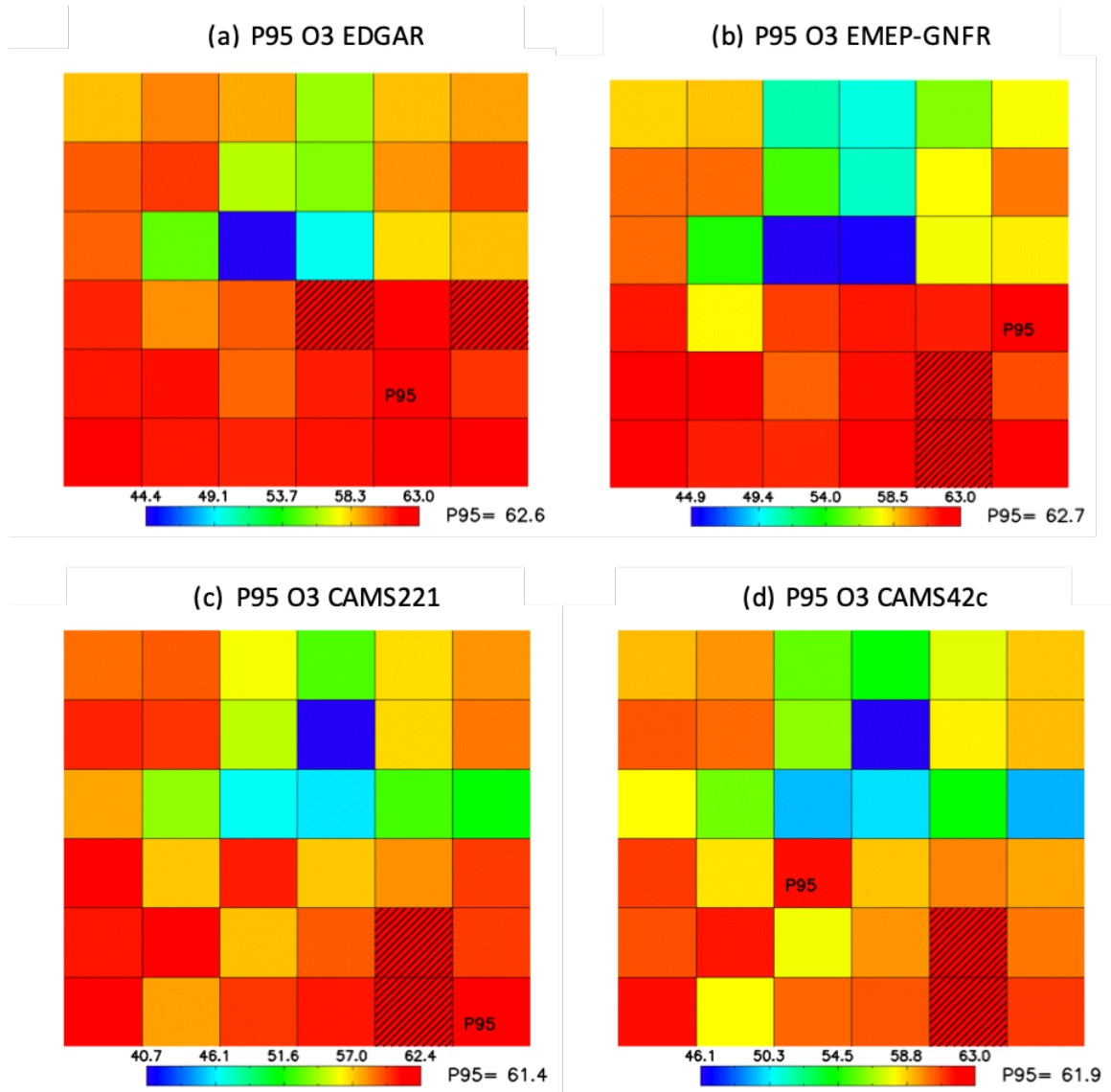

Figure 8. Overview of the location of the P95 values for the calculated O3 concentrations µg/m3 by the four Base Cases for the domain BRU. Shaded grid cells indicate the location of values above the P95 values by (a) EDGAR, (b) EMEP-GNFR, (c) CAMS221 and (d) EMEP42C. The number next to P95 represents the average of the P95 values.


Table 8(a) Overview of BaseCase emissions (mg/m2/day) for NOx and VOC, together with the ratio in the emissions between these two pollutants. (b) Similar as to (a) but for potency at P95 in mg/m3.


(a)

| Emissions [mg/m2/day] | | | | | Ratio Emissions VOC/NOx | | | |
|---|---|---|---|---|---|---|---|---|
| NOx | EDGAR | EMEP-GNFR | CAMS221 | CAMS42C | EDGAR | EMEP-GNFR | CAMS221 | CAMS42C |
| BER | 9.76 | 8.50 | 7.72 | 8.24 | 1.60 | 0.90 | 1.11 | 1.07 |
| BRU3 | 31.57 | 27.21 | 31.05 | 28.16 | 1.28 | 0.56 | 0.52 | 0.53 |

| | | | | | | | | |
|-----|-------|-------|-------|-------|------|------|------|------|
| BUC | 13.94 | 17.01 | 19.93 | 18.28 | 1.89 | 0.92 | 1.41 | 1.17 |
| MAD | 16.15 | 16.59 | 19.44 | 15.05 | 1.32 | 1.14 | 1.04 | 1.26 |
| MAL | 7.36 | 7.79 | 8.20 | 8.04 | 1.65 | 0.86 | 0.78 | 0.88 |
| POV | 5.23 | 5.62 | 5.48 | 5.16 | 1.27 | 1.20 | 1.11 | 1.11 |
| ROM | 12.01 | 11.54 | 12.26 | 10.11 | 1.75 | 1.44 | 1.44 | 1.58 |
| STO | 10.71 | 7.41 | 7.01 | 6.03 | 1.69 | 1.03 | 1.38 | 1.57 |

**Emissions [mg/m2/day]**

| VOC | EDGAR | EMEP-GNFR | CAMS221 | CAMS42C |
|------|-------|-----------|---------|---------|
| BER | 15.65 | 7.67 | 8.55 | 8.78 |
| BRU3 | 40.50 | 15.29 | 16.13 | 14.93 |
| BUC | 26.35 | 15.73 | 28.15 | 21.40 |
| MAD | 21.28 | 18.97 | 20.31 | 19.00 |
| MAL | 12.11 | 6.71 | 6.35 | 7.06 |
| POV | 6.65 | 6.72 | 6.08 | 5.74 |
| ROM | 21.05 | 16.66 | 17.69 | 16.02 |
| STO | 18.13 | 7.63 | 9.69 | 9.46 |

**(b)**

| | Potency P95 (μg/m3/ton) | | | | Ratio Potency VOC/NOx | | | |
|---------|--------|-----------|---------|---------|-------|-----------|---------|---------|
| NOx 50% | EDGAR | EMEP-GNFR | CAMS221 | CAMS42C | EDGAR | EMEP-GNFR | CAMS221 | CAMS42C |
| BER | 0.011 | 0.011 | 0.007 | 0.008 | -0.27 | -0.27 | -0.43 | -0.38 |
| BRU3 | 0.063 | 0.051 | 0.040 | 0.073 | -0.06 | -0.12 | -0.13 | -0.08 |
| BUC | 0.041 | 0.066 | 0.029 | 0.031 | -0.32 | -0.26 | -0.34 | -0.42 |
| MAD | 0.013 | 0.016 | 0.014 | 0.017 | -0.31 | -0.19 | -0.29 | -0.18 |
| MAL | 0.000 | 0.000 | 0.001 | 0.000 | - | - | -1.00 | - |
| POV | -0.003 | -0.002 | -0.001 | -0.003 | 0.33 | 0.50 | 1.00 | 0.67 |
| ROM | 0.044 | 0.051 | 0.052 | 0.046 | -0.41 | -0.41 | -0.40 | -0.46 |
| STO | 0.013 | 0.014 | 0.012 | 0.013 | -0.15 | -0.21 | -0.17 | -0.15 |


| Potency P95 (μg/m3/ton) | | | | |
|-------------------------|--------|-----------|---------|---------|
| VOC 50% | EDGAR | EMEP-GNFR | CAMS221 | CAMS42C |
| BER | -0.003 | -0.003 | -0.003 | -0.003 |
| BRU3 | -0.004 | -0.006 | -0.005 | -0.006 |
| BUC | -0.013 | -0.017 | -0.010 | -0.013 |
| MAD | -0.004 | -0.003 | -0.004 | -0.003 |
| MAL | -0.001 | -0.001 | -0.001 | -0.001 |
| POV | -0.001 | -0.001 | -0.001 | -0.002 |
| ROM | -0.018 | -0.021 | -0.021 | -0.021 |
| STO | -0.002 | -0.003 | -0.002 | -0.002 |

**Table 9. Absolute potential (50%) divided by the Absolute potential (25%) for O3 when NOx emissions are reduced by 50% and 25% for 95 Percentile values (P95). Numbers with more than 5% non-linearity are highlighted.**

| City | EDGAR | EMEP-GNFR | CAMS221 | CAMS42C |
|------|-------|-----------|---------|---------|
| BER | 0.94 | 0.94 | 0.90 | 0.91 |
| BRU3 | 1.00 | 1.00 | 1.00 | 1.01 |
| BUC | 0.90 | 0.93 | 0.92 | 0.92 |
| MAD | 0.87 | 0.88 | 0.88 | 0.88 |
| MAL | 12.03 | 0.25 | 0.73 | 5.38 |
| POV | 1.37 | 1.54 | 1.58 | 1.41 |

| | | | | |
|---|---|---|---|---|
| ROM | 0.91 | 0.93 | 0.93 | 0.92 |
| STO | 0.93 | 0.92 | 0.91 | 0.91 |

**Table 10. Absolute potential (50%) divided by the Absolute potential (25%) for O3 when VOC emissions are reduced by 50% and 25% for 95 Percentile values (P95).**

| City | EDGAR | EMEP-GNFR | CAMS221 | CAMS42C |
|---|---|---|---|---|
| BER | 1.00 | 1.00 | 1.01 | 1.01 |
| BRU3 | 1.00 | 1.01 | 1.01 | 1.01 |
| BUC | 1.01 | 1.01 | 1.01 | 1.01 |
| MAD | 0.99 | 1.00 | 0.99 | 0.99 |
| MAL | 1.02 | 1.01 | 1.01 | 1.02 |
| POV | 1.04 | 1.02 | 1.03 | 1.03 |
| ROM | 1.01 | 1.00 | 1.00 | 1.01 |
| STO | 1.00 | 1.00 | 1.01 | 1.01 |


**Table 11. Absolute potential (50%) divided by the Absolute potential (25%) for O3 when NOx and VOC emissions are reduced together by 50% and 25% for 95 Percentile values (P95). Numbers with more than 5% non-linearity are highlighted.**

| City | EDGAR | EMEP-GNFR | CAMS221 | CAMS42C |
|---|---|---|---|---|
| BER | 0.96 | -1.40 | 0.90 | 0.92 |
| BRU3 | 1.00 | 2.35 | 1.00 | 1.01 |
| BUC | 0.92 | 0.76 | 0.94 | 0.94 |
| MAD | 0.87 | 0.69 | 0.88 | 0.89 |
| MAL | 1.63 | 0.40 | 0.74 | 1.46 |
| POV | 1.17 | 1.57 | 1.19 | 1.17 |
| ROM | 0.91 | 0.86 | 0.94 | 0.92 |
| STO | 0.94 | 0.77 | 0.91 | 0.91 |
