# Peer review of "Sensitivity of air quality indicators to emission inventories (EDGAR, EMEP, CAMS-REG) in Europe through FAIRMODE benchmarking methodology."

_EGUsphere, 2023_

## Author Comment (AC1)

Thank you for your constructive suggestions which have been very helpful in improving the manuscript. We have been keen to follow them up. Please find below a point-to-point reply to the comments.

**Reviewer 2**

*This study compares four different emission inventories in their impact on air quality indicators derived from the FAIRMODE project focusing on PM10 and O3 for the year 2015. The air quality indicators compare the effect of emission perturbations on air pollutant concentrations for the different emission inventories and eight cities / regions in Europe. The study is understood as an expansion of the work by Thunis et al. (2022). The overall structure of the article is good and generally well written. The results are presented in a consistent way. However, I would appreciate a more in-depth discussion of the results. The used methodology illustrates inconsistencies between different emission inventories, which are well identified. The authors lack to put these inconsistencies into context and to find suggestions for updates of the inventories. Further, the article lacks to place the proposed work into the context of recent work on emission inventory intercomparisons/evaluations, especially from the work that is published by the authors themselves, which does require some expansion of the introduction. Given the importance of anthropogenic emissions on air quality forecast and their intrinsic uncertainty, this study is relevant for publication. The study fits to ACP in the field of air quality modelling evaluation.*

*General comments:*

*- The main objective of the analysis and the main conclusions drawn are not clear. I understand that the objective of the article is to compare the emission inventories and their effect on the evaluation of emission reductions. But it is not clear to me what the authors conclude from the comparison. What conclusions can by drawn for updating the emission inventories? What are suggestions for further air quality evaluations or comparisons of emission inventories?*
We inserted lines at the end of the conclusions section to stress the actions that should be considered to increase the robustness of the emission inventories.
"From an emission inventory viewpoint, this work indicates that the most efficient actions to improve the robustness of the modelling responses to emission changes would be to better assess the sectorial share and total quantities of PPM emissions. Another important aspect is to better assess emitted precursor ratios as these lead to important differences in model responses, both in the case of O3 (NOx/VOC ratio) and PM (NOx/NH3/SO2 ratios).
From a modelling point of view, NOx responses are the more challenging and require caution because of their non-linearity.
We also added at several places in the manuscript elements to better stress the outcome of the comparison and how this outcome may impact emission development work.

*- The title highlights the sensitivity of air quality indicators to emission inventories. I do understand that the indicators differ for different emission data, but the conclusion drawn from this is not clear to me. This needs more elaboration throughout the article. Further, the*

*focus of the article is on the response of the EMEP model to changes in the emission inventories, rather than on the inventories themselves. This is not clear from the title. I recommend to update the title in this regard.*

We have changed the title to better reflect the focus of our work. It now reads as: "Sensitivity of air quality model responses to emission changes: comparison of results based on four EU inventories through FAIRMODE benchmarking methodology.

*- The abstract needs some update. More quantitative results need to be given. Also, the logical structure of the abstract needs some revision. The results are not presented in a logical ordering.*

We replaced the abstract with the following text.

Despite the application of an increasingly strict EU air quality legislation, air quality remains problematic in large parts of Europe. To support the abatement of these remaining problems, a better understanding of the potential impacts of emission abatement measures on air quality is required and air chemistry transport models (CTMs) are the main instrument to perform emission reduction scenarios. In this study, we study the robustness of the model responses to emission reductions when emission input is changed. We investigate how inconsistencies in emissions impact the modelling responses in the case of emission reduction scenarios. Based on EMEP simulations over Europe fed by four emission inventories: EDGAR 5.0, EMEP-GNFR, CAMS 2.2.1 and CAMS version 4.2 (incl. condensables), we reduce anthropogenic emissions in six cities (Brussels, Madrid, Rome, Bucharest, Berlin and Stockholm) and 2 regions (Po Valley Italy and Malopolska Poland) and study the variability of the concentration reductions obtained with these four emission inventories.

Our study reveals that the impact of reducing aerosol precursors on PM10 concentrations result in different potentials and potencies, differences that are mainly explained by differences in emission quantities, differences in their spatial distributions as well as in their sector allocation. In general, the variability among models is larger for concentration changes (potentials) than for absolute concentrations. Similar total precursor emissions can however hide large variations in sectorial allocation that can lead to large impacts on potency given their different vertical distribution. PPM appears to be the precursor leading to the major differences in terms of potentials. From an emission inventory viewpoint, this work indicates that the most efficient actions to improve the robustness of the modelling responses to emission changes would be to better assess the sectorial share and total quantities of PPM emissions. From a modelling point of view, NOx responses are the more challenging and require caution because of their non-linearity. For O3, we find the relationship between emission reduction and O3 concentration change shows the largest non-linearity for NOx (concentration increase) and a quasi-linear behaviour for VOC (concentration decrease).

We also emphasize the importance of accurate ratios of emitted precursors since these lead to changes of chemical regimes, directly affecting the responses of O3 or PM10 concentrations to emission reductions.

*- The experimental setup should be made more clear. Which simulations have been performed (1 basecase, 2 scenarios for each city and precursor?).*

Done. We added the following in section 2:

"Four emission inventories are used to feed the EMEP model to understand how this input data influences the calculated model changes in air pollutant concentrations. We performed one BaseCase simulation with each emission inventory for the year 2015 over Europe.

For the scenarios, we reduced for each emission inventory, the emissions of NOx, VOCs, NH3, SOx and primary particulate matter (PPM which includes both their fine (size <2.5 μm) and coarse (2.5 μm< size <10 μm) by 25% and 50% for each species separately. This is done for six cities (Brussels, Madrid, Rome, Bucharest, Berlin and Stockholm) and two regions (Malopolska, Poland and Po Valley, Italy) to study the impact on particulate matter (PM) and ozone (O3) formation.

*- The naming of the different emission inventories is not consistent. Please revise the article carefully and rephrase. Please consider to not use EMPE, EMEPG …. Instead, use the original name of the emission inventory (e.g., similar to Fig. S2).*
We corrected the names of the four emission inventories in the manuscript as suggested by the reviewer. Also, all the tables and figures have been adapted.

*- The "non-linearites" sections (3.4, and 3.7) should be sub-subsections (3.3.6 and 3.6.3) to illustrate that they belong to PM and O3 discussions, respectively.*
We agree with the reviewer that the sections on non-linearities could be sub-sections to PM and O3 respectively and adapted the manuscript accordingly.

*- The article discusses several times that the P95 grid cells are not similar for the different emission inventories, which makes the comparison biased. Please redo the calculations for the same grid cell for all inventories, by using either only those grid cell that are commonly exceeding the 95th percentile or by using all grid cells for which at least one emission inventory produces exceeding values.*
We use P95 cells that are consistent between the BaseCase and scenarios for each model. But they can differ from one model to the other. We do not see this as a bias because we are interested to compare models for the highest concentration range. If fixing similar cells for all models, we would introduce another type of bias as the comparison would then mix high and low concentration cells, probably characterised by different chemical and physical processes.

*- I appreciate the bullet points in the final section giving main findings. However, the findings are general and usually well known. It is not clear, what specific recommendations the authors conclude from the results that could help improving emission inventories.*
We added a paragraph after the bullet points to summarize what practical actions could be taken to improve the consistency of the emission inventories.

*- All figures should carefully be checked. Most figures show a gray shadow and subplot indicators (a,b,c…) are partly overlayed by the subplot. Please also increase the resolution of the figures.*
Corrected.

*- "Basecase" is not spelled consistently throughout the article. Please revise carefully and choose one spelling, also in the formulars.*
We went through the manuscript, and made changes accordingly.

*- In the introduction there is a tendency to highlight the uncertainty of emission inventories. However, this should be more elaborated in the results/conclusion section*
We emphasized in the manuscript and especially in the conclusion that additional efforts to check the consistency and accuracy of the PPM emissions and their sectorial share is therefore important to ensure robust model responses.

*- The introduction needs more guidance toward the importance of this study. E.g., why do we need emission reduction scenario? Also, a short review of what is done in the literature so far is missing completely. How can the study complement current knowledge.*
Thank you for your suggestion. We added the following text to the Introduction:
"Many studies exist that analyse the sensitivity of baseline concentrations to emissions or have compared model responses among themselves (Thunis et al., 2007, 2010, 2013, 2021a Vautard et al., 2007, Mircea et al., 2019). To the knowledge of the authors, very few works assessed the sensitivity of model responses to the emission input, e.g. De Meij et al. (2009), Aman et al., (2011) Miranda et al. (2015 and references therein)."

Furthemore, we changed the text in the Introduction to emphasize the need of emission reduction studies.
"While in Thunis et al. (2022), the authors compared emission inventories among themselves and proposed an approach to identify inconsistencies, we here investigate how these inconsistencies impact the modelling responses in the case of emission reduction scenarios. It is indeed crucial to better assess the share of the uncertainty that is associated to emission inventories in the overall uncertainty of the modelling response (Georgiou et al., 2020) as this is a key model output when designing air quality plans."

Thunis, P., Rouil, L., Cuvelier, C., Stern, R., Kerschbaumer, A., Bessagnet, B., Schaap, M., Builtjes, P., Tarrason, L., Douros, J., Moussiopoulos, N., Pirovano, G., Bedogni, M.,: Analysis of Model Responses to Emission-reduction Scenarios within the CityDelta Project. ATMOSPHERIC ENVIRONMENT 41; 10.1016/j.atmosenv.2006.09.001, 2007.

Vautard, R., Builtjes, P.H.J., Thunis, P., Cuvelier, C., Bedogni, M., Bessagnet, B., Honoré, C., Moussiopoulos, N., Pirovano, G., Schaap, M., Stern, R., Tarrason, L., Wind, P.,: Evaluation and intercomparison of Ozone and PM10 simulations by several chemistry transport models over four European cities within the CityDelta project, Atm. Env., 41, https://doi.org/10.1016/j.atmosenv.2006.07.039, 2007.

Thunis, P., Cuvelier, C., Roberts, P., White, L., Nyrni, A., Stern, R., Kerschbaumer, A., Bessagnet, B., Bergstrom, R., Schaap, M.,: EURODELTA - Evaluation of a Sectoral Approach to Integrated

Assessment Modeling - Second Report. EUR 24474 EN. Luxembourg (Luxembourg): Publications Office of the European Union; 10.2788/40803, 2010

Mircea, M., Bessagnet, B., D'Isidoro, M., Pirovano, G., Aksoyoglu, S., Ciarelli, G., Tsyro, S., Manders, A., Bieser, J., Stern, R., García Vivanco, M., Cuvelier, C., Aas, W., Prévôt, A.S.H., Aulinger, A., Briganti, G., Calori, G., Cappelletti, A., Colette, A., Couvidat, F., Fagerli, H., Finardi, S., Kranenburg, R., Rouïl, L., Silibello, C., Spindler, G., Poulain, L., Herrmann, H., Jimenez, J.L., Day, D.A., Tiitta, P., Carbone, S.: EURODELTA III exercise: An evaluation of air quality models' capacity to reproduce the carbonaceous aerosol, Atm.c Env. X, 2, https://doi.org/10.1016/j.aeaoa.2019.100018, 2019.

Thunis, P., Pernigotti, D., Cuvelier, C., Georgieva, E., Gsella, A., De Meij, A. Pirovano G., Balzarini, A., Riva, G. M., Carnevale, C., Pisoni, E., Volta, M., Bessagnet, B., Kerschbaumer, A., Viaene, P., De Ridder, K., Nyiri, A., and Wind, P., POMI: A Model Intercomparison exercise over the Po valley, Air Quality, Atmosphere and health, DOI: 10.1007/s11869-013-0211-1, October 2013.

Amann, M., Bertok, I., Borken-Kleefeld, J., Cofala, J., Heyes, C., Höglund-Isaksson, L., Klimont, Z., Nguyen, B., Posch, M., Rafaj, P., Sandler, R., Schöpp, W., Wagner, F., Winiwarter, W.,: Cost-effective control of air quality and greenhouse gases in Europe: Modeling and policy applications, Env. Mod. Soft., 26, https://doi.org/10.1016/j.envsoft.2011.07.012, 2011.

De Meij, A., et al.,: The sensitivity of the CHIMERE model to emissions reduction scenarios on air quality in Northern Italy, Atmos. Env., doi:10.1016/j.atmosenv.2008.12.036, 2009c.

Miranda, A., Silveira, C., Ferreira, J., Monteiro, A., Lopes, D., Relvas, H., Borrego, C., Roebeling, P.,: Current air quality plans in Europe designed to support air quality management policies, Atmos. Poll. Res., 6, 2015, https://doi.org/10.5094/APR.2015.048.

*Specific comment:*

*- line 36-38: This statement is vague and cannot be concluded from the article. It needs a reference in the discussion or an evaluation in the results sections. As the title indicates a focus on the air quality indicators, the analysis on how the chosen metric (mean or P95 values) affect the analysis is appreciated.*
The reviewer has a good point. We added to the text (in section 3.3.7): "This corroborates the results by Thunis et al., (2021c), who assessed the contribution of cities to their own air pollution. They showed that the type of indicator impacts the final outcome, i.e. the share of the city pollution caused by their own emissions in his study. It also confirms that indicators based on averaged values tend to report more linear relationships."

Thunis, P., Clappier, A., de Meij, A., Pisoni, E., Bessagnet, B., and Tarrason, L.: Why is the city's responsibility for its air pollution often underestimated? A focus on PM2.5, Atmos. Chem. Phys., 21, 18195–18212, https://doi.org/10.5194/acp-21-18195-2021, 2021c.

*- line 50: Start the sentence with "Among others, air chemistry …" as only two options of the model applications are given.*
Done.

*- line 51: "where no observations are available": More over, CTM are used to complement observations also where observations are present. Please rephrase.*
Done, thank you.

*- Paragraph starting at line 56: This can be more concise. There are a number of repetitive statements, first in general, and another time in the frame of FAIRMODE.*
We have made corrections to the text.

*- line 78: Sentence starting with "Followed by the analysis" is not complete. Please revise.*
Corrected.

*- line 79: add a "," after "In chapter 4"*
Done.

*- line 83: Remove "," after "EMEP model".*
Done.

*- line 84: At this point of the article, it is not clear what "model responses" mean, response to what effect?*
We changed that into "model responses (in terms of concentration) to emission reductions.

*- line 86: Please take the study of Clappier et al. (2021) for the effect of emission reductions on air pollution concentrations. They choose similar emission reductions and discussed the non-linearity associated with the reductions. doi: 10.1016/j.envint.2021.106699*
In our study we compare four different emission inventories, while Clappier et al., (2021) studies the impact of $NO_x$, $SO_2$ and $NH_3$ reductions on EU scale on secondary inorganic PM2.5 concentrations. We believe that making a reference to Clappier et al., 2021 does not fit in Section 2 Methodology. Yet, we think that it is more appropriate to make reference to Clappier et al., (2021) in section 3 Results, to compare our results with theirs.

*- line 87: Not clear, why the authors take these cities / regions into consideration. Is there a motivation? Why not considering London or Paris instead of Stockholm? Also, given a rather coarse model resolution for cities scale air quality evaluation, is it really advisable to focus on air quality in cities. I assume a larger error than considering regions (as you did with the Po Valley and Malopolska). And explanatory sentence is appreciated.*

The choice of the cities is quite arbitrary and results from different considerations. First these cities were studied in other exercises like FAIRMODE and future comparison with these is seen as a possible future activity. Second, we tried to identify cities where pollution is a key issue (Stockholm would in this respect be less interesting). Finally, we must stress that emission reductions are applied in one single EU simulation. Cities and regions must therefore be far away from each other to avoid that reductions applied over one city or region potentially influence the background concentration levels in another city/region. This constraint limits the number of cities/regions that we can manage in this work. We added this to the text in section 2.

*- line 89-90: Please rephrase the sentence: "Note that the analysis for Malopolska and the Po Valley is exclusively for the city centre of Krakow and Milan, respectively, while the emission reductions act on the whole regions, as described…" or similar.*

Corrected. The sentence now reads as: "For Malopolska and the Po Valley emissions are reduced over the whole modelling domain, as described in Table S1 of the Electronic Supplement. However, we analyse the impact of the emission reductions only over the city centres of Krakow and Milan, respectively."

*- line 93: Please add the information that the emissions are reduced in a rather large area encompassing the different cities (for Rome this is approx. 100 km x 100 km).*

We have made it clearer in the text in section 2.

*- Section 2.2.2: How are the EMEP model and EMEP-GNFR emissions related? Are they interlinked or does the names solely link to the UNECE programme?*

We rephrased the sentence. "The EMEP emissions are provided by GNFR (Gridded Nomenclature For Reporting) sector."

*- line 144: It is not clear what the bracket term "(Member states, in Europe)" mean. Please rephrase.*

We removed that.

*- line 152: change ", Granier" in "(Granier"*

Done.

*- line 152/159: Name the emitted species directly at the beginning instead of "main air pollutants and greenhouse gases"*
Done.

*- line 166: What is CAMS-REF1? Is it the CAMS inventory described above? Please revise the article carefully and name the inventories concisely.*
We modified the sentence and renamed the inventories concisely through the text.

*- line 171/172: Move to the end of section 2.2, as they are general for all inventories.*
Done.

*- line 186: definition of "Baseline" is missing. Please indicate that this is the emission inventories data. Further, define the scenarios explicitly as 25% and 50% emission reduction for each pollutant.*
Corrected. The sentence now reads as:
"$C_{BaseCase}$ represents the BaseCase yearly concentrations, obtained with one of the four emission inventories (no emission reduction)."
In line with one of the previous comments by the reviewer, we have provided a better description of the BaseCase and the different scenarios, in section 2.

*- line 187: Remove "etc." as there are no more scenarios in this article.*
Done.

*- line 189/190: Please rephrase: "Note that the grid cells exceeding the 95th percentile are always with respect to the baseline simulation." or equivalent.*
Corrected.

*- line 190/191: "projected to 100%": This is not clear, given the non-linearities that are likely for high emission reductions. Please rephrase.*
The definition of the potential includes the scaling. Therefore, we have modified the sentence.
"The absolute potential informs on the concentration change projected **linearly** to 100% from a given scenario".

*- line 212: Please revise the article carefully: The use of (p,s) as pollutant/sector pair is not relevant for this study. It is more appropriate to use precursor/city (p,c) or alike. This should be changed throughout the article.*
We agree with the reviewer and changed the text of the entire section so that it is now referring to precursor and city. Thank you.

*- line 214: Please remove "(1)"*
Done.

*- line 214: Please make clear that E is the emission difference between the scenario and the inventory for the equations to hold true.*
E is here intended as the absolute emission values. Multiplied by alpha, we then obtain the emission reduction change, i.e. deltaE=alpha*E. We clarified this in the text

*- line 218: Remove the extra bracket*
Done.

*- line 230: Please rephrase the sentence starting with "It will be..".*
Corrected. "This is the case."

*- line 234: Please rephrase the sentence starting with "In practice". I understand that the AP for each city needs to exceed 20 % of the maximum AP calculated among all cities. This is not clear from this sentence and formula.*
We modified the sentence as follows: This is achieved by imposing that any given potential fulfils the condition: $AP_{p,c} > \gamma \times \max\{AP_{p,c}\}$ to be further considered in the screening, where $\gamma$ is a user defined threshold parameter, set to 20% in this work.

*- line 253: The diagram shows shapes for different cities, not sectors. Please revise.*
Corrected. The sentence now reads as "In this diagram, shapes are used to differentiate precursors while colours differentiate cities."

*- line 259/260: Please indicate, which emission inventories are exactly used to calculate the median (as this is done further below in the article)*
Done.

*- line 315: Add a full stop "." after "Thunis et al. (2022)"*
Done.

*- line 323: The first two paragraphs of this section are general knowledge and can be removed. This is not the scope of the article.*
Done.

*- line 349: Please add: "… calculated PM10 concentrations of" a 50 % "emission reduction…"*
Done.

*- line 354: Please add: "… a factor of two and five" difference "as compared to …"*
Done.

*- line 374: Please add: "This explains the much higher potencies in EMEG" for Stockholm.*
Done.

*- line 375: Please add "among all precursors" to the end of the sentence.*
Done.

*- line 376: Remove the first "is".*
Done.

*- line 377/378: The same information as in the sentence before. Please remove.*
Done.

*- line 391/392: Please rephrase the sentence. There are to many nouns.*
Corrected. The sentence now reads as "In the next section we analyse the impact of aerosol secondary precursors reductions on calculated PM10 concentrations."

*- line 410/411: I can't find the differences for Bucharest and Malopolska in the plot. Do you mean Berlin and Madrid? Please revise.*
We corrected the sentence. "For MAD and BRU, we see that higher SOx emissions (factor ~2) by EDGAR are compensated by lower potencies, which lead to overall similar potentials. Hence, reducing SOx emissions in EDGAR has a larger impact on PM10 concentrations when compared to the median, via the chemical reactions that lead to the formation of ammonium sulphate aerosol as described in De Meij et al. (2009c)."
Thank you.

*- line 412: EMEPE has a larger impact on PM10 compared to which inventory/median. Please revise.*
Corrected; "when compared to the Median", see previous comment.

*- line 417: Please change "problem" to "inconsistency".*
Done.

*- line 429: This paragraph is not linked to VOC, thus, it should start a new section. The same holds true for the next paragraph. However, the sections in this part of the article are very short. Please consider merging them into a single section with appropriate headline.*

Reviewer 1 had a similar suggestion. Therefore, we have created a new paragraph 3.3.6 that contains the analysis on the SOx/NOx ratios.

*- line 438: Remove the second "emissions".*

Done.

*- line 448: Please add the corresponding table (Nox/NH3 ratios) to the supplement material.*

We added the following tables to the Supplement material:

**Table S4(a) Overview of Base Case emissions (mg/m2/day) for NOx and NH3x, together with the ratio in the emissions between these two pollutants. (b) Similar as to (a) but for potency at P95 in µg/m3.**

(a)

| Emissions mg/m2/day | | | | | Ratio emissions NOx/NH3 | | | |
|---|---|---|---|---|---|---|---|---|
| NOx | EDGAR | EMEP-GNFR | CAMS221 | CAMS42C | EDGAR | EMEP-GNFR | CAMS221 | CAMS42C |
| BER | 9.764 | 8.501 | 7.717 | 8.236 | 2.43 | 3.76 | 2.05 | 2.53 |
| BRU3 | 31.572 | 27.208 | 31.046 | 28.161 | 4.10 | 8.16 | 5.79 | 5.52 |
| BUC | 13.944 | 17.014 | 19.926 | 18.279 | 4.18 | 3.86 | 2.89 | 2.73 |
| MAD | 16.149 | 16.586 | 19.437 | 15.053 | 4.99 | 8.70 | 6.21 | 4.18 |
| MAL | 7.36 | 7.79 | 8.195 | 8.039 | 2.09 | 6.14 | 4.57 | 3.93 |
| POV | 5.229 | 5.617 | 5.482 | 5.155 | 1.14 | 1.78 | 1.69 | 1.63 |
| ROM | 12.012 | 11.544 | 12.257 | 10.11 | 3.29 | 4.80 | 4.86 | 5.39 |
| STO | 10.709 | 7.411 | 7.01 | 6.027 | 6.42 | 12.22 | 6.62 | 10.03 |

| Emissions mg/m2/day | | | | |
|---|---|---|---|---|
| NH₃ | EDGAR | EMEP-GNFR | CAMS221 | CAMS42C |
| BER | 4.023 | 2.261 | 3.766 | 3.253 |
| BRU3 | 7.694 | 3.335 | 5.362 | 5.106 |
| BUC | 3.336 | 4.404 | 6.886 | 6.69 |
| MAD | 3.238 | 1.907 | 3.13 | 3.598 |
| MAL | 3.522 | 1.269 | 1.792 | 2.046 |
| POV | 4.583 | 3.156 | 3.235 | 3.172 |
| ROM | 3.652 | 2.406 | 2.524 | 1.875 |
| STO | 1.668 | 0.6067 | 1.059 | 0.6006 |

(b)

| Potency P95 (µg/m³/ton) | | | | | Ratio Potency NOx/NH3 | | | |
|---|---|---|---|---|---|---|---|---|
| NOx | EDGAR | EMEP-GNFR | CAMS221 | CAMS42C | EDGAR | EMEP-GNFR | CAMS221 | CAMS42C |
| BER | -0.0018 | -0.0011 | -0.0024 | -0.0018 | 0.11 | 0.07 | 0.19 | 0.14 |
| BRU3 | 0.0013 | 0.0013 | 0.0015 | 0.0012 | -0.05 | -0.02 | -0.03 | -0.02 |
| BUC | -0.0067 | -0.0047 | -0.0019 | -0.0017 | 0.07 | 0.12 | 0.05 | 0.05 |
| MAD | 0.0002 | -0.0002 | -0.0002 | -0.0013 | -0.01 | 0.01 | 0.01 | 0.07 |
| MAL | -0.001 | -0.0011 | -0.0004 | -0.001 | 0.20 | 0.09 | 0.05 | 0.10 |
| POV | -0.0064 | -0.0047 | -0.0046 | -0.0059 | 1.64 | 0.67 | 0.79 | 0.78 |
| ROM | -0.022 | -0.0076 | -0.0151 | -0.0089 | 0.27 | 0.10 | 0.17 | 0.10 |
| STO | -0.0011 | -0.0011 | -0.0006 | -0.0005 | 0.01 | 0.02 | 0.01 | 0.01 |

| Potency P95 (µg/m³/ton) | | | | |
|---|---|---|---|---|
| NH₃ | EDGAR | EMEP-GNFR | CAMS221 | CAMS42C |
| BER | -0.0162 | -0.0168 | -0.0125 | -0.013 |
| BRU3 | -0.0253 | -0.0738 | -0.0473 | -0.0507 |
| BUC | -0.095 | -0.0384 | -0.0398 | -0.0375 |
| MAD | -0.0229 | -0.0227 | -0.0193 | -0.0174 |
| MAL | -0.0051 | -0.0116 | -0.0082 | -0.0098 |
| POV | -0.0039 | -0.007 | -0.0058 | -0.0076 |
| ROM | -0.0803 | -0.0773 | -0.0884 | -0.0928 |
| STO | -0.0779 | -0.0644 | -0.0515 | -0.049 |

*- line 454: NOx- vs. NH3 sensitivity depends on the relative share. If no NH3 is available to form PM, adding more NOx does not lead to more PM. Conversely, if enough NOx is available to form PM even after reducing NOx emissions, the emission reductions do not have a positive impact on PM reductions. Please rephrase the sentence on NOx-sensitivity.*
Corrected.

This section now reads as "While NOx emissions in the four inventories are similar, EDGAR contains almost a factor 2 more NH3 emissions. This means that NH3 is relatively more abundant in EDGAR and its reduction has therefore less impact on concentartion. This results in the formation processes being more 'NOx-sensitive' in Rome. Thus, reducing NOx in EDGAR leads to larger impact on PM10 concentrations."

*- line 468: Please rephrase: What does "indicating a larger efficiency for more important emission reductions" mean?*
This is addressed in the next comment.

*- line 469/470: Please rephrase the sentence, e.g.. "indicating that a larger NOx emission reduction becomes more efficient in reducing PM concentrations" or equivalent.*
The sentence now reads as:

"For NOx (Table 3) the behaviour is generally non-linear with ratios larger than 1.00. This indicates that calculated PM10 concentrations would be more reduced between 25 and 50% than between 0 and 25%."

*- line 482: "less important": I would consider this a similar magnitude of non-linearity given the values in the tables. Please rephrase.*
We removed "although less important than for NOx".

*- line 483: What does "more important" mean? Please rephrase.*
Corrected. It now read as "emissions are reduced further in a NH3-limited regime".

*- line 486: Consider removing "dilution" by "compensating".*
As suggested by the reviewer, we replaced "dilution" by "compensating".
Reviewer 1 suggested to replace "NOx non-linearities are diluted by other emitted species", by "NOx non-linearities are weakened by other emitted species".

*- line 488: Please revise carefully: Probably Malopolska should be BUC?*
Corrected, thank you.

*- line 514: Please add: "… leading to similar potentials. This is not the …" (making two sentences out of one)*
Done.

*- line 520: Is "important the correct wording? Please rephrase. Further, please change "many cities" to "three cities"*
Corrected.

*- line 530: Please remove: "… leading to NOx symbols…"*
Done.

*- line 545: Change "number" to "value"*
Done.

*- line 533: Please add: "…VOC/NOx" ratio.*
Done.

*- line 554: Please rephrase "twice larger" to "twice as large" or "two times larger"*
Corrected.

*- line 572: Please rephrase the sentence starting with "A number larger than 1…": Superlinearity is the term for this effect.*
Thank you for this. The sentence now reads as:
"A number larger than 1 indicates superlinearity; that means that O3 concentrations are more reduced between 25 and 50% than between 0 and 25%."

*- line 603: remove extra "s"*
Done.

*- line 608: please add: "concentration) especially" for "PPM."*
Done.

*- line 631/632: As the title indicates, the evaluation of the behavior of air quality indicators is key in the article. Given this, I would have expected an evaluation of different choices of indicator. Please add this to the article or change the title accordingly.*
We have changed the title, as mentioned earlier.

*- line 806: Please remove the "emissions" at the end of the line*
Done, thank you.

*- Figure 3/7: The plots are very busy due to the dashed lines. Please consider reducing the number of dashed lines, increase the figures resolution, do not overlay the figure annotation (e.g., fac2) and the axes. Carefully check all the figures to avoid overlaying the subfigure ID (i.e., (a), (b)…) by the subplot*
Corrected.

*- 839: please add: "EMEPC42C (green)" for each GNFR sector.*
Done.

*- line 843/844: Please rephrase the first sentence of the figure's caption.*
Corrected. The first sentence now reads as: "Overview of the location of the P95 values for the calculated PM10 concentrations µg/m3 by the four Base Cases for the domain STO."

*- line 849: Change 95P → P95*
We went through the text and replaced 95P to P95. Thank you.

*- Table 2-10: Please change the city IDs according to table S1 in the supplement.*
We have changed all city IDS in all the tables accordingly. Thank you.

*- line 896: Please rephrase the first sentence of the figure's caption*
The first sentence of the figure caption now reads as: Figure 8. Overview of the location of the P95 values for the calculated O3 concentrations µg/m3 by the four Base Cases for the domain BRU.

*- Table 7b: The VOC/NOx ratio is missing. Please add this*
Corrected. Thank you.

*Supplement material:*

*- Figure S1: Please consider to plot the emission per grid cell and avoid contour plots. The red rectangle is missing for Malopolska and the Po valley. If emission reductions act on the whole domain shown in the subplots, please add this information to the caption.*
Reviewer 1 has a similar comment. We corrected all the maps in Fig. S1 of the ES and added to the text that for the Po Valley and Malopolska the emissions are reduced over the entire domain.
Below we show for illustration purposes only the new Figures S1 ac, ad, ae, and af.

**PM25 total emissions CAMS221**
Po Valley mg/m2

[Figure]

**PM25 total emissions CAMS42C**
Po Valley mg/m2

(ac)                                                    (ad)

**PM25 total emissions EDGAR**
Po Valley mg/m2

[Figure]

**PM25 total emissions EMEP-GNFR**
Po Valley mg/m2

(ae)                                                    (af)

---

## Author Comment (AC2)

Thank you for your constructive suggestions which have been very helpful in improving the manuscript. We have been keen to follow them up. Please find below a point-to-point reply to the comments.

**Reviewer 1**

General Comments:

*The authors conduct year-long model simulations over Europe with the EMEP CTM to intercompare four emission inventories in order to capture the uncertainty in FAIRMODE air quality metrics for PM and Ozone, focusing on emission control scenarios on specific urban centres and conglomerates. This is timely given the recent update of WHO guideline values, as well as the EU policy targets for improving air quality, and relevant within the scope of ACP (Methods for assessment of models).*

*The language throughout the manuscript should be improved to be made more fluent and precise, and avoid repetition.*
We went through the manuscript carefully, removed repetitions and improved the text where possible.

*In particular the abstract should be re-written to more clearly state the objectives and outcomes, avoid repetitions and include quantitative as well as qualitative comparisons.*
We replaced the abstract with the following text.
Despite the application of an increasingly strict EU air quality legislation, air quality remains problematic in large parts of Europe. To support the abatement of these remaining problems, a better understanding of the potential impacts of emission abatement measures on air quality is required and air chemistry transport models (CTMs) are the main instrument to perform emission reduction scenarios. In this study, we study the robustness of the model responses to emission reductions when emission input is changed. We investigate how inconsistencies in emissions impact the modelling responses in the case of emission reduction scenarios. Based on EMEP simulations over Europe fed by four emission inventories: EDGAR 5.0, EMEP-GNFR, CAMS 2.2.1 and CAMS version 4.2 (incl. condensables), we reduce anthropogenic emissions in six cities (Brussels, Madrid, Rome, Bucharest, Berlin and Stockholm) and 2 regions (Po Valley Italy and Malopolska Poland) and study the variability of the concentration reductions obtained with these four emission inventories. Our study reveals that the impact of reducing aerosol precursors on PM10 concentrations result in different potentials and potencies, differences that are mainly explained by differences in emission quantities, differences in their spatial distributions as well as in their sector allocation. In general, the variability among models is larger for concentration changes (potentials) than for absolute concentrations. Similar total precursor emissions can however hide large variations in sectorial allocation that can lead to large impacts on potency given their different vertical distribution. PPM appears to be the precursor leading to the major differences in terms of potentials. From an emission inventory viewpoint, this work indicates that the most efficient actions to improve the robustness of the modelling responses to emission changes would be to better assess the sectorial share and total quantities of PPM emissions. From a modelling point of view, NOx responses are the more challenging and require caution because of their non-linearity. For O3, we find the relationship between emission reduction and O3 concentration change shows the largest non-linearity for NOx (concentration increase) and a quasi-linear behaviour for VOC (concentration decrease).

We also emphasize the importance of accurate ratios of emitted precursors since these lead to changes of chemical regimes, directly affecting the responses of O3 or PM10 concentrations to emission reductions.

*The authors should make clear how this study is related and complementary to Thunis et al. (2022) where emission inventories for 150 cities are investigated?*
In Thunis et al. (2022), inconsistencies were only analysed at the level of the emission inventory. In this work we go one step further by assessing how these emission inconsistencies impact the model responses to emission reduction scenarios. We check whether differences in model results arise mainly from inconsistencies in emission (input data) or from the model itself. We also introduced the concept of the ensemble (median) to facilitate the comparison.
We added this to the manuscript in section 2.

*Specific Comments:*

*In Sec. 2.2 it would be good for the reader if a table is added summarising the species present in each inventory and highlighting the differences, e.g. resolutions, bottom-up/country-totals methodology, compilation year etc.*

We added the following to the manuscript:
Table 1. Overview of the four emission inventories used in this study.

| Name inventory | Resolution (lon x lat) in degrees | Method | Release date | Sector classification | Condensables included | Total NOx emissions* | Total SOx emissions* | Total PM25 emissions* | Total NH3 emissions* |
|---|---|---|---|---|---|---|---|---|---|
| Edgar_v5.0 | 0.1 x 0.1 | Bottom-up | 2020 | 13 GNFR | No | 6360 | 4074 | 1278 | 5116 |
| EMEP | 0.1 x 0.1 | Country report | 2018 | 13 GNFR | No | 7445 | 2591 | 1229 | 3663 |
| CAMS 2.2.1 | 0.1 x 0.05 | Country report | 2018 | 13 GNFR | No | 6410 | 2513 | 1272 | 3708 |
| CAMS4.2C | 0.1 x 0.05 | Country report | 2022 | 12 GNFR | Yes | 6419 | 2519 | 1688 | 3640 |

*Total emissions for Austria, Belgium, Bulgaria, Denmark, Finland, France, Greece, Hungary, Ireland, Italy, Luxembourg, Netherlands, Poland, Portugal, Romania, Spain, Sweden, Estonia, Latvia, Lithuania, Czech Republic, Slowakia, Slovenia ,Croatia, Cyprus, Malta and Germany in Ktons/year.
The anthropogenic emissions in the four inventories are: CO, NOx, SOx, NH3, VOC, PM25, PM10. Edgar uses a bottom-up approach for all emission source sectors, based on estimates of activity data and emission factors whereas CAMS is mainly based on countries reported emissions. The differences between the same years between the CAMS inventories stems from the recalculations of the pollutants for each country.

*In 2.2.4 could you please make more clear in the text how the "condensables" are different to previously reported PM2.5/10 and to quantify the expected differences, also for the examples of Poland and Turkey?*
We changed the text of section 2.2.4 to emphasize the impact of condensables with respect to previously reported emissions. The text now reads as:
This inventory (Kuenen et al., 2021, 2022) is an update of the previous CAMS versions for PM emissions for the residential sector, also known as REF1, in which PM2.5 and PM10 emissions have

been updated with information on the condensable part (personal communication J. Kuenen, TNO, 2021). This inventory, also known as REF2, is hereafter denoted as CAMS42C. Condensables replace country reported PM2.5 and PM10, with a bottom-up estimate for small combustion for all fuels (not only wood but also for fossil fuels). Since 2016, more and more countries gradually included condensable emissions of small combustion devices, leading to significant differences as shown by Kuenen et al. (2022). For example, in countries such as Poland and Turkey where coal combustion in households is still an important contributor to PM, large emissions of fine and coarse condensables (118kTons/year for PM25) still take place. For Turkey the difference in PM2.5 emissions for GNFR Sector 03 is around 20% (higher in CAMS42C). For Hungary, Slovakia, Ireland, UK, Belgium and Norway the PM2.5 emissions for GNFR Sector 03 are in general lower than in CAMS42C.

The figure below (not added to the manuscript) shows the differences in PM2.5 emissions for Sector 3 (Domestic heating) between CAMS42C and CAMSv211. Over Poland, France, Slovenia and parts of the Netherlands we see large differences between the two inventories. For example, for Poland CAMS42C has about three times more PM2.5 emissions (~118kTons/year) than CAMS221. For Turkey the difference in PM2.5 emissions for GNFR Sector 03 is around 20% (higher in CAMS42C). For Hungary, Slovakia, Ireland, UK, Belgium and Norway the PM2.5 emissions for sector 3 are in general lower in CAMS42C, as illustrated in the figure below.

**DIFF PM25_sec03 CAMS42C vs CAMS221**

[Figure]

[Figure]

| PM25 Sec03 (tons/year) | CAMS 2.2.1 | CAMS42+C |
|---|---|---|
| Turkey | 120893 | 145168 |
| Poland | 65421 | 184363 |

*In Sec. 2.4 the screening method should be better motivated. With which criteria were the user-defined thresholds decided? The first 3 paragraphs read as a user guide for the FAIRMODE output diagram in Fig. 3 rather than a discussion on statistical screening - please consider rephrasing.*
The thresholds are arbitrary but should ideally be chosen in such a way to distinguish uncertainties from inconsistencies. Too small differences will not allow to distinguish between inconsistency and uncertainty whereas large enough differences must be considered as inconsistencies. It must be noted that these thresholds can be lowered as the analysis proceeds and inconsistencies are

progressively solved or explained. We added some sentences in the manuscript to explain these points. Regarding the 3 first paragraphs of Section 2.4, we believe they are necessary to understand the basic principles underlying our approach.

*Sec. 3.1: What were the selection criteria for the cities, regions included in the study? Would it be possible to include additional locations (even to expand to all major European cities) to make the study even more encompassing and robust? (For example in Thunis et al. (2022) emission inventories for 150 cities are investigated - can this be done here or in a subsequent study to statistically assess the effectiveness and impacts of EU-wide measures?)*

The impact of emission reductions on concentrations is calculated in one single simulation for all cities/regions. The areas where emission reductions are applied are therefore selected in such a way that they are far away from each other to avoid that reductions applied over one area influence the background concentration levels in another, which would hamper our analysis. In this context, extending the analysis to 150 cities is not possible. We added this to the text in section 2.

*Sec. 3.2: Are the aerosol processes (secondary production) reported included and captured by the CTM model used? Are natural aerosols such as dust included in the modelled PM10?*

Secondary aerosol formation is included in the EMEP model, together with biogenic VOC and Dimethyl Sulphide (DMS) emissions, together with natural dust (e.g. windblown dust from deserts, semis-arid areas, agricultural and bare lands, and Saharan; Simpson et al. 2012).

We added to Section 2.1 that secondary aerosol formation in the EMEP model is included. We removed the first two paragraphs of section 3.2 "Variability of PM10 .." as suggested by Reviewer 2.

*Sec. 3.5 can be merged with Sec. 3.6: the ratio of NOx/VOC is important - are the urban centres studied here NOx/VOC limited in terms of O3 production/sink? Is there a different seasonal dependence in the model results from city to city that would be important for air quality plans?*

We disagree with the reviewer regarding Section 3.5 and 3.6 that deal with different aspects. On the other hand, we agree that seasonal temporal variations in the emission inventories are important on gas and aerosol calculations as shown in De Meij et al. 2009. Clappier et al. (2021) analysed the seasonal variation of the chemical regimes all over Europe. We've added the following to the text "Clappier et al. (2021) showed which chemical regimes are responsible to the secondary inorganic PM formation over Europe, and how these chemical regimes can help in designing efficient PM abatement strategies. They showed that during wintertime, PM25 concentrations are predominantly NH3-sensitive in the major part Europe. During summertime, PM25 are predominantly SO2-sensitive in most of Europe."

Regarding the reviewer's question on the dependence from city to city, in general, city centres are VOC-limited due to the abundance of NOx emissions caused by road transport. Clappier et al., (2021) showed that for VOC-limited O3 formation regime areas, where NOx emission reductions of 50% lead to substantial increases in O3 in wintertime due to a decreased titration of O3 by NO Clappier et al. (2021).

Furthermore, the seasonal and geographical dependency on O3 formation/depletion is addressed in a joint paper in FAIRMODE (accepted in Journal Air Quality, Atmosphere & Health), that describes

the impact of short-term emission reductions on the calculated O3 concentrations for different cities in Europe.

*Sec. 3.7: Given the difference in behaviour in specific regions how are the results of this study to be interpreted? Can the EMEP model provide a map of the different chemical regimes across European cities/regions for each inventory? What is the non-linear behaviour regarding reduction of both NOx and VOC?*

Thanks for pointing out this issue. We agree with the reviewer that chemical regimes can greatly vary between seasons and between different locations in Europe. The non-linear behaviour on O3 formation in Europe is studied by Beekmann and Vautard (2010), who provide a comprehensive study on O3 formation chemical regimes over Europe. They showed that during summer time, VOC-limited regimes are present especially over urbanized areas while NOx sensitive chemical regimes occur over southern Europe.

Reducing $NO_x$ and VOC emissions together also shows the non-linear behaviour when $NO_x$ and VOC emissions are reduced together by different quantities, see Table 11. The formation of $O_3$ is less sensitive to the reduction of $NO_x$ emissions when simultaneously also VOC emissions are reduced. This corroborates the findings of Xiao et al., 2010, Xing et al., 2017. We have made it clearer in the text.

In addition to O3, Clappier et al. (2021) showed which chemical regimes are responsible to the secondary inorganic PM formation over Europe, and how these chemical regimes can help in designing efficient PM abatement strategies. They showed that during wintertime, PM25 concentrations are predominantly NH3-sensitive in the major part Europe. During summertime, PM25 are predominantly SO2-sensitive in most of Europe.

Thunis et. al (2021) showed that the peculiarity of secondary PM2.5 formation in the Po basin, which is characterised by contrasting chemical regimes within distances of a few (hundred) kilometres, as well as non-linear responses to emission reductions during wintertime.

Beekmann, M. and Vautard, R.: A modelling study of photochemical regimes over Europe: robustness and variability, Atmos. Chem. Phys., 10, https://doi.org/10067-10084, 2010.

A. Clappier, P. Thunis, M. Beekmann, J.P. Putaud, A. de Meij, Impact of SOx, NOx and NH3 emission reductions on PM2.5 concentrations across Europe: Hints for future measure development, Environment International, Volume 156, 2021, ISSN 0160-4120, https://doi.org/10.1016/j.envint.2021.106699.

Thunis, P., Clappier, A., Beekmann, M., Putaud, J. P., Cuvelier, C., Madrazo, J., and de Meij, A.: Non-linear response of PM2.5 to changes in NOx and NH3 emissions in the Po basin (Italy): consequences for air quality plans, Atmos. Chem. Phys., 21, 9309–9327, https://doi.org/10.5194/acp-21-9309-2021, 2021.

*Sec.4 Please consider using paragraphs rather than bullet points to present the findings.*

We decided to keep bullet points to improve readability of the main findings but added a final concluding paragraph. Note that the bullet points were appreciated by Reviewer 2. We however added a final concluding paragraph after the bullet points.

*Please clearly explain early in the text how is PMcoarse different to PM10? What is PPM and how is PPM10 and PPM2.5 different to PM10 and PM2.5? Please make sure all acronyms are properly defined and used.*

Particles between 2.5 and 10 micrometers in diameter are referred to as PM coarse particles. PPM10 and PPM2.5 contain the primary component of the PM10 or PM2.5 fractions respectively. This is different from PM10, which contains also the secondary part (inorganic and organic aerosols such as sulphates, nitrates, ammonium and biogenic organic aerosols). We added this in Section 2 as suggested by the reviewer. We also checked that all acronyms are defined and consistent.

*P95 should be defined when first used.*

Done.

*Technical Corrections: (page, line number):*
*p3l53 newer, better -> more elaborate*
Done. We added to the sentence: "contributing to smaller biases when compared to observations".

*p3l56 uncertainties associated to certain processes -> associated with*
Corrected.

*p3l63 One of FAIRMODE's goal -> goals*
Corrected.

*p3l64 explain -> investigate*
Corrected.

*p4l121 What does it mean "only differ in terms of version"? Please clarify.*
Corrected. It now reads as "release date and emission updates".

*p13l434 Start a new subsection so it's clear the discussion is not about VOC.*
Corrected. We did the same in the section for O3 (now 3.6.3)

*p13l440: Can you explain why NOx has to compete with NH3 to form PM?*
We rephrased the sentence and now read as:
The reduction in NO2 concentrations leads to a reduction in HNO3 while the increase in oxidant concentrations increases the formation of HNO3. These two competing mechanisms effect the production of nitrate aerosol via HNO3 + NH3.

*p14l486: Please consider the use of a different word than dilution which has a specific meaning in chemistry that might confuse the reader (ie. rephrase "diluted by other emitted species")*
Corrected and replaced by "weakened".

*For the plots in the supplement pages 3-10 it would be advisable to use raster graphics showing the output on the native grid, rather than interpolated values on the map.*

As suggested by the reviewer we corrected all the maps in Fig. S1. Below we show for illustration purposes only the new Figures S1 ac, ad, ae, and af.

**PM25 total emissions CAMS221**
Po Valley mg/m2

[Figure]

**PM25 total emissions CAMS42C**
Po Valley mg/m2

(ac)

(ad)

**PM25 total emissions EDGAR**
Po Valley mg/m2

[Figure]

**PM25 total emissions EMEP-GNFR**
Po Valley mg/m2

(ae)

(af)